*Resource*

# Ontology-guided clustering enables proteomic analysis of rare pediatric disorders

Ericka C M Itang [1,2], Vincent Albrecht[1], Alicia-Sophie Schebesta[1,2], Marvin Thielert [1],
Anna-Lisa Lanz [3], Katharina Danhauser[3], Jessica Jin[3], Tobias Prell[3], Sophie Strobel[3],
Christoph Klein [2,3], Matthias Mann [1], Susanne Pangratz-Fuehrer[2,3] & Johannes Mueller-Reif [1,2]✉

## Abstract

**The study of rare pediatric disorders is fundamentally limited by small patient numbers, making it challenging to draw meaningful biological conclusions. To address this, we developed a framework integrating clinical ontologies with proteomic profiling, enabling the systematic analysis of rare conditions in aggregate. We applied this approach to urine and plasma samples from 1140 children and adolescents, encompassing 394 distinct disease conditions and healthy controls. Using advanced mass spectrometry workflows, we quantified over 5000 proteins in urine, 900 in undepleted (neat) plasma, and 1900 in perchloric acid-depleted plasma. Embedding SNOMED CT clinical terminology in a network structure allowed us to group rare conditions based on their clinical relationships, enabling statistical analysis even for diseases with as few as two patients. This approach revealed molecular signatures across developmental stages and disease clusters while accounting for age- and sex-specific variation. Our framework provides a generalizable solution for studying heterogeneous patient populations where traditional case-control studies are impractical, bridging the gap between clinical classification and molecular profiling of rare diseases.**

**Keywords** Pediatrics; Plasma; Proteomics; SNOMED CT; Urine
**Subject Categories** Genetics, Gene Therapy & Genetic Disease; Methods & Resources; Proteomics

## Introduction

The study of rare pediatric disorders presents a fundamental challenge in biomedical research: despite their collective impact on child health, individual conditions often affect too few patients for traditional analytical approaches. This challenge is particularly acute in molecular profiling studies, where small patient cohorts limit statistical power and make it difficult to draw meaningful conclusions about disease mechanisms. While previous research has successfully characterized individual pediatric conditions such as cancer (Lorentzian et al, 2023), cardiovascular diseases (Goldenberg et al, 2014), and inflammatory disorders (Nygaard et al, 2024), the molecular understanding of hundreds of rare pediatric diseases remains limited. These limitations underscore the need for integrative frameworks that can leverage clinical and molecular data to uncover shared mechanisms across rare diseases, even when individual patient numbers are small.

Network-based approaches in precision medicine have been pivotal in addressing such challenges. Clinical ontologies such as the Systematized Nomenclature of Medicine Clinical Terms (SNOMED CT) organize medical knowledge into structured hierarchies, enabling the systematic grouping of related conditions and providing a foundation for analyzing diseases in aggregate (Cornet and De Keizer, 2008; El-Sappagh et al, 2018; Chang and Mostafa, 2021). Similarly, molecular network analyses have proven invaluable for integrating omics data to uncover shared mechanisms across diseases (Buphamalai et al, 2021; Jagtap et al, 2022). Knowledge graph-based approaches have further demonstrated the potential of linking clinical and proteomics data in precision medicine (Santos et al, 2022). Despite these advancements, the integration of clinical ontologies like SNOMED CT with proteomics analysis remains largely unexplored. Such a combination offers the potential to create a unified framework that captures both clinical relationships and molecular patterns across rare and common pediatric disorders.

Quantitative proteomics enhances this approach by capturing disease-specific molecular signatures alongside developmental and demographic influences. Several advancements in mass spectrometry (MS) technologies have enabled the robust quantification of thousands of proteins across a wide dynamic range. High-throughput MS workflows, including the dia-PASEF (data-independent acquisition parallel accumulation—serial fragmentation) acquisition mode (Meier et al, 2020) and multiplexed approaches such as the mTRAQ-based plexDIA (Derks et al, 2023) and the dimethyl-based multiplexed-DIA (Thielert et al, 2023), have significantly improved the throughput of proteomic analyses without sacrificing depth and reproducibility. Beyond these technological developments, proteomics has been increasingly

[1]Department of Proteomics and Signal Transduction, Max Planck Institute of Biochemistry, Martinsried, Germany. [2]German Center for Child and Adolescent Health (DZKJ), partner site Munich, Munich, Germany. [3]Department of Pediatrics, Dr. von Hauner Children's Hospital, Ludwig-Maximilians-Universität München, Munich, Germany.
✉E-mail: jomueller@biochem.mpg.de

applied in clinical contexts, with body fluids serving as accessible windows into the health status of patients (Guo et al, 2025). In blood plasma proteomics, enrichment and depletion strategies have been developed to extend proteome coverage, specifically for low-abundance proteins (Viode et al, 2023; Albrecht et al, 2025). These innovations now allow for the quantification of thousands of proteins in clinical samples, making them particularly suitable for studying the molecular diversity present in pediatric cohorts. For instance, MS-based proteomics has been used to investigate changes in the proteome during growth, development, and puberty, shedding light on how developmental transitions and hormonal fluctuations shape proteomic variability (Niu et al, 2025; Bjelosevic et al, 2017).

Here, we present a novel framework that combines clinical ontologies with large-scale proteomics to study rare pediatric disorders. We analyzed urine and plasma proteomes from 1140 children and adolescents with 394 distinct conditions, leveraging both biofluids to provide complementary insights into systemic and local disease mechanisms. Urine and plasma were selected for their minimally invasive collection, making them especially suitable for pediatric patients. To increase our plasma proteome coverage, we complemented undepleted plasma proteomics (neat plasma) with a perchloric acid depletion (PCA-N plasma) strategy (Albrecht et al, 2025), a streamlined adaptation of the perCA workflow developed by the Hanno Steen group (Viode et al, 2023). This approach depletes highly abundant plasma proteins and enhances the detection of low-abundance targets. By integrating an ontological network based on SNOMED CT clinical terminology with proteomic data, we systematically grouped related rare diseases, enabling the study of conditions with very small patient numbers by leveraging their relationships within the broader landscape of pediatric disorders. Through comprehensive profiling of both urine and plasma proteomes, combined with precise integration of demographic and clinical data, our approach creates a foundation for understanding the molecular basis of pediatric diseases that have historically been difficult to study in isolation.

# Results

## Overview of the pediatric cohort

We studied a cohort of 1140 children and adolescents, aged 3–17 years, from the Dr. von Hauner Children's Hospital in Munich (Fig. 1A). This cohort consisted of 131 healthy individuals and 1009 patients diagnosed with a range of 394 distinct pediatric disease conditions (Fig. 1B). Among these, 859 participants provided both urine and plasma samples, while 141 provided only urine samples and 140 only plasma samples (Fig. 1C). Dataset EV1 summarizes the baseline characteristics of this pediatric cohort, for which we performed MS-based proteomic profiling on both urine and plasma samples (Fig. 1D,E and "Methods").

## Proteome data quality assessment

We assessed the quality and reproducibility of our proteomic measurements separately for urine and plasma. For the urine proteome, we used a multiplexed data-independent acquisition (mDIA) workflow with a pooled reference channel, employing isotopically distinct dimethyl mass tags (light Δ0, intermediate Δ4, and heavy Δ8) for differential sample labeling. The reference channel, composed of pooled urine peptides from all individuals in the cohort and labeled with the light (Δ0) tag, provided a consistent standard across all multiplexed samples. This workflow achieved excellent labeling efficiency and maintained a false discovery rate below 1% through optimized filtering parameters (Fig. EV1A–C). Analysis of over 1000 urine samples demonstrated robust reproducibility, with a median analytical coefficient of variation (CV) of 21% (Fig. EV1D).

For the plasma proteome, we employed a label-free data-independent acquisition (DIA) workflow, using quality control (QC) samples composed of pooled plasma peptides from several individuals in the cohort. QC measurements in the neat plasma dataset showed highly reproducible protein group quantifications, with a median analytical CV of 13% across all measurements (Fig. EV1E). In contrast, QC measurements in the PCA-N plasma dataset showed higher proteomic depth (1217 protein group identifications compared to 658 in neat plasma) but at the cost of increased variability, with an analytical CV of 29%, which agrees with published investigation about the effects of the PCA-N method (Albrecht et al, 2025).

In brief, PCA-N-depleted proteins do have a higher variance but, at the same time, greater overlap to the urine proteome. Therefore, we prioritized the neat plasma dataset for global comparisons, including comparisons between urine and plasma proteomes and age- and sex-dependent proteome variability. However, for disease-related changes, where detecting low-abundance proteins is particularly relevant, we integrated the depleted plasma dataset on top of the neat plasma dataset to enhance our sensitivity to biologically relevant disease markers.

## Comparing the urine and plasma proteome of pediatric individuals

To pre-process the urine cohort dataset, we applied stringent filtering to exclude 280 samples with low proteomic depth (<800 protein group identifications). We then performed batch correction using pyCombat to mitigate batch effects associated with the sample plate. This approach identified a cumulative total of 5081 protein groups, of which 41.5% (2109 protein groups) had a biological CV below 50% (Fig. EV2A). On average, 1589 protein groups were quantified per urine sample, with protein intensities spanning five orders of magnitude (Fig. EV2B). In addition, 20.5% of the identifications (1041 protein groups) contained more than 60% valid values (Fig. EV2C). The relatively low data completeness and high biological CV in the urine dataset reflect the high interindividual variability in protein composition among developing children, likely influenced by factors such as hydration, diet, and activity level.

For the neat plasma cohort dataset, we pre-processed the data by excluding 10 outlier samples with protein group identifications below the lower threshold (calculated as the 25th percentile minus 1.5 times the interquartile range). We then performed batch correction using pyCombat to address sample plate-associated batch effects. This preprocessing step identified a cumulative total of 920 protein groups, with 91.4% (841 protein groups) exhibiting a biological CV below 50% (Fig. EV2A). On average, 632 protein groups were quantified per plasma sample, with protein intensities

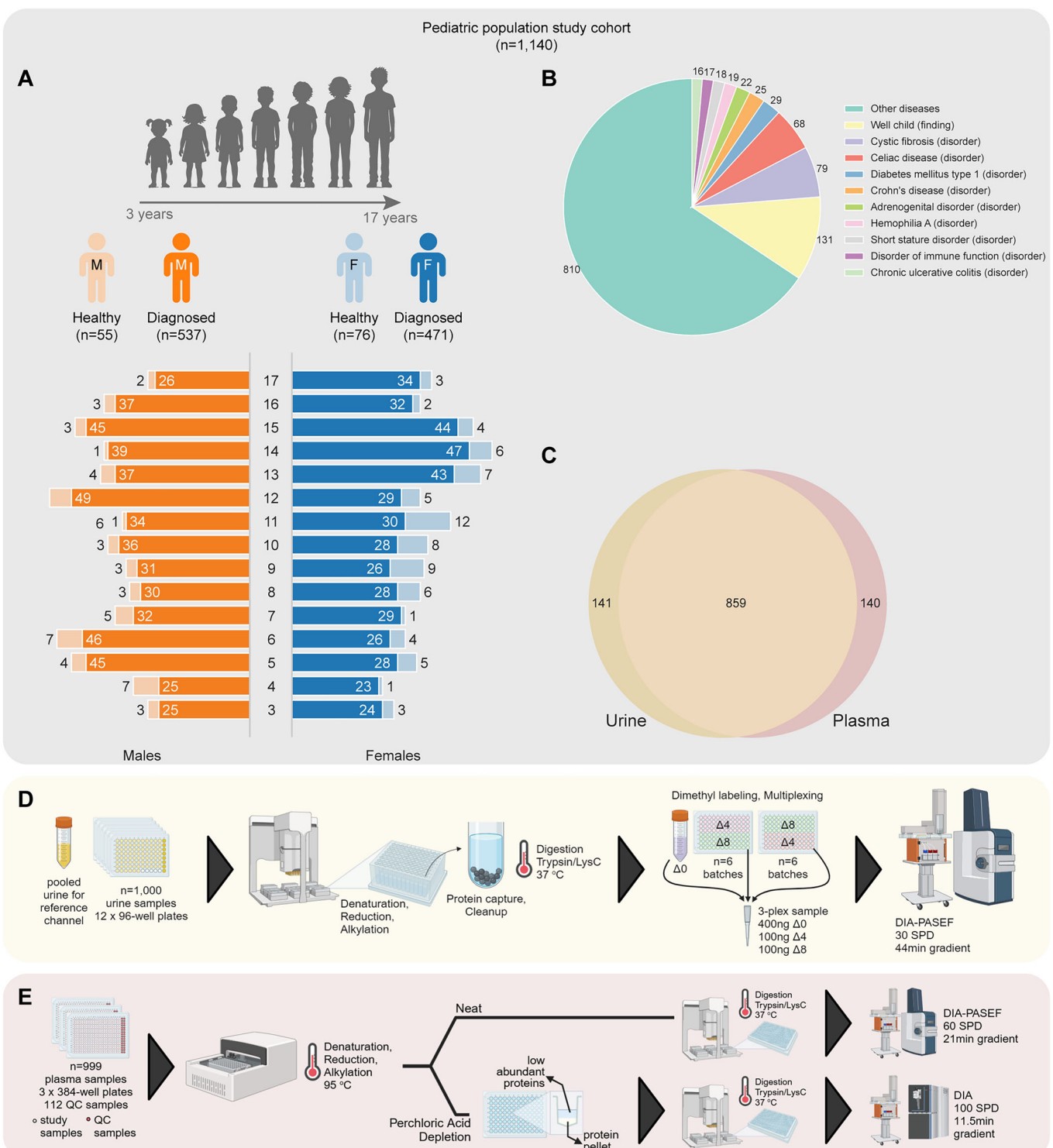

**Figure 1. Overview of the cohort and the proteomics workflow.**

(A) Composition of the study cohort, characterized by biological sex, age, and group (control or diagnosed). (B) Distribution of the pediatric disorders within the study cohort, illustrating the top 10 most frequent diagnostic categories, including the "Well child (finding)" label for the control group. (C) Venn diagram showing participants who provided urine samples, plasma samples, or both. (D) Workflow for urine proteome profiling using the dimethyl-based multiplexed-DIA (mDIA) approach. Created with BioRender.com. (E) Workflow for plasma proteome profiling using the label-free DIA approach. Both neat and PCA-N plasma were analyzed. The depletion strategy was conducted using perchloric acid to selectively precipitate out high-abundance proteins. Created with BioRender.com.

spanning six orders of magnitude (Fig. EV2B). Notably, the plasma dataset demonstrated higher data completeness than the urine dataset, with 66.6% of identifications (613 protein groups) containing more than 60% valid values (Fig. EV2C).

In total, 485 protein groups were commonly identified in urine and plasma (Fig. EV3A). Their median intensities showed a weak positive correlation (Pearson's $r = 0.29$), forming a distinct plasma-predominant cluster and a urine-predominant cluster (Fig. EV3B).

Proteins in the plasma-predominant cluster primarily included albumin, transferrin, and proteins involved in lipid transport, such as several apolipoproteins (APOA1, APOA2, APOB, APOC3, APOD), as well as proteins related to coagulation and fibrinolysis (F2, FGA, FGB, FGG, PLG, SERPINC1). These proteins are expected to be highly abundant in plasma due to their essential roles in active systemic functions within the circulatory system (Benjamin and McLaughlin, 2012; Havel, 1987; Risman et al, 2023). Their presence in plasma but limited detection in urine indicates that they are generally retained in the bloodstream rather than filtered through the kidneys.

In contrast, the urine-predominant cluster featured proteins associated with kidney function, filtration, and epithelial protection (e.g., AMBP, B2M, and TIMP1), along with structural proteins like keratins (KRT1, KRT2, KRT5) and mucins (MUC1) that may have been shed into urine due to epithelial cell renewal and turnover in the renal and urinary tract lining (Pastushkova et al, 2016; Bragulla and Homberger, 2009).

We further analyzed the Pearson correlations between urine and plasma protein intensities for each individual protein. This revealed that many proteins with moderate Pearson correlation scores (Pearson's $r \geq 0.4$ and FDR-adjusted $P$ value < 0.01) were associated with immunological functions (Fig. EV3C). These included the variable chains of immunoglobulins (IGKV1D-13, IGHV3-64, IGHV1-3, IGLV10-54, IGHV2-70), as well as the immune-regulatory molecule FGL1 that functions as a major inhibitory ligand for LAG-3, thus modulating T-cell mediated immune responses (Wang et al, 2019). This suggests immune activity in plasma, with several immune complexes or fragments passing into urine through filtration mechanisms. The urine-plasma proteome correlation plots for the moderately correlating proteins mentioned above are shown in Fig. EV3D.

## Effects of age and biological sex on the urine and plasma proteome

Age is a significant factor influencing the human proteome. Prior studies have established that protein expression in body fluids varies considerably with age. This is especially true in children and adolescents, as rapid physiological and developmental changes drive dynamic fluctuations in protein expression (Niu et al, 2025; Lietzén et al, 2018; Bjelosevic et al, 2017). Our data confirms these age-dependent patterns in both the urine and plasma proteome of developing children.

Principal component analysis (PCA) revealed that although there are no distinct clusters associated with specific age groups, there is a continuous gradient along PC1 that correlates with increasing age (Fig. 2A,B, left panel). Further analysis of PC1 against age showed that PC1 values rise more sharply after age 10–12, presumably reflecting the significant hormonal and developmental changes characteristic of pubertal onset (Fig. 2A,B, right panel).

To further explore age-dependent proteomic changes, we performed one-way analysis of variance (ANOVA) of the protein intensities across different biological sexes and age groups and unsupervised hierarchical clustering analysis on the z-scored mean protein intensities of the significantly different proteins (FDR-adjusted $P$ value < 0.01) across age and sex (Dataset EV2). This analysis revealed five main modules of distinct protein abundance trajectories for each of the urine proteome (Fig. 2C) and the plasma proteome (Fig. 2D). Notably, we observed greater variance in protein abundance trajectories across ages between males and females in the urine proteome compared to the plasma proteome, where the trajectory across ages remains consistent for both sexes. This is particularly evident in module 1 of the urine proteome, where there is a sharp increase in protein intensities after age 9 in females, in contrast to a more gradual increase in males. Further exploration of this module indicates a strong enrichment, based on combined scores, of proteins associated with Gene Ontology (GO) terms related to biological processes such as "keratinocyte differentiation" and "autocrine signaling". The enrichment of these terms suggests significant roles for these processes in puberty-related physiological changes, presumably reflecting differences in skin maturation and hormone-driven signaling pathways between males and females. Keratinocyte differentiation is critical for skin barrier formation and maintenance, and this process is known to be modulated by sex hormones such as androgens and estrogens (Zouboulis and Degitz, 2004; Zouboulis, 2009). Autocrine signaling, which involves cells responding to their own secreted signals, may modulate local tissue responses and hormonal feedback loops, influencing sex-specific development (Sato, 1999; Prange-Kiel et al, 2003). The largest module identified in the urine proteome, Module 5, comprises 170 proteins and exhibits a decreasing trend in protein abundance with increasing age. This module is primarily associated with Gene Ontology (GO) terms such as "elastic fiber assembly" and "regulation of extracellular matrix disassembly," suggesting an age-related decline in structural protein activities and ECM interactions, potentially reflecting changes in tissue remodeling and processes like skin aging, which are characterized by reduced elasticity and altered extracellular matrix composition (Li et al, 2021; McCabe et al, 2020). Modules 2 to 4 show erratic trajectories that could possibly be an artifact of the high interindividual variability in the urine proteome.

In the plasma proteome, Module 4, as the largest module, comprises 100 proteins and shows a linear positive correlation between protein intensity and age for both males and females. GO term enrichment analysis of this module revealed that the most enriched terms are "B-cell receptor signaling pathway" and "positive regulation of respiratory burst", likely reflecting the maturation of the adaptive immune system during this developmental window. Module 3 displays a linear relationship as well, but with protein intensities decreasing with age for both sexes. This module is enriched for GO terms such as "complement activation, lectin pathway" and "positive regulation of opsonization", reflecting a downregulation of proteins involved in innate immune mechanisms. Notably, low-abundance plasma proteins involved in cytokine activity, such as the monocyte differentiation antigen (CD14), the cell-surface receptor (CD44), and the macrophage colony-stimulating factor 1 receptor (CSF1R), are included in this module. These proteins play key roles in innate immune responses and demonstrate the capability of our workflow to detect biologically

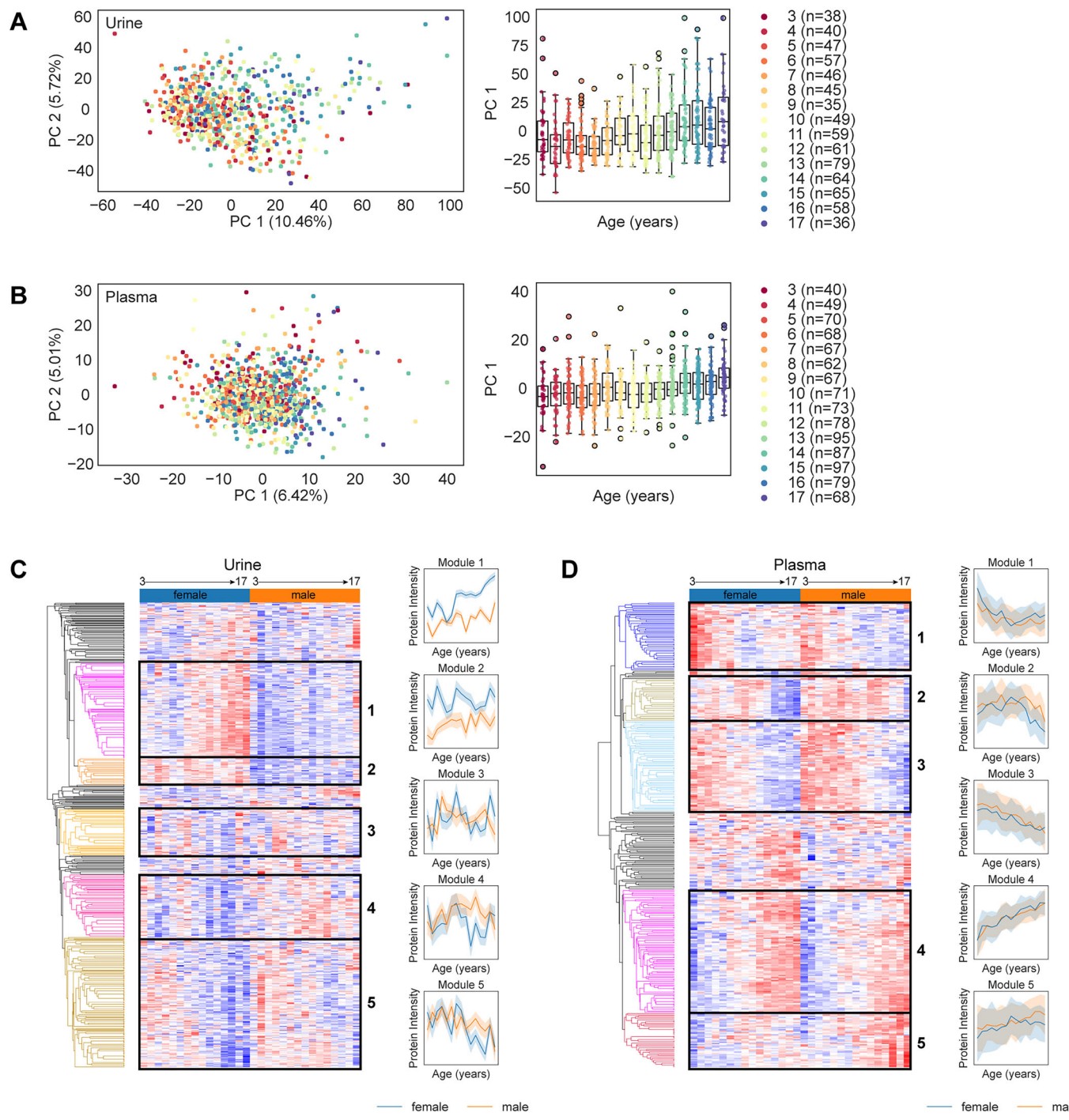

**Figure 2. Dependency of the proteome profile on age and biological sex.**

(A) Left panel: Distribution of urine samples across the first two principal components, with each dot representing an individual sample, color-coded by age in years. Right panel: Distribution of the first principal component (PC1) across different age groups shown as boxplots with overlaid jitter plots. The boxes represent the interquartile range (IQR; 25th to 75th percentiles), with the median value indicated by a central line. Whiskers extend to the minimum and maximum values within 1.5 times the IQR from the box. A legend on the right specifies the number of samples per age group. (B) Left panel: Distribution of neat plasma samples across the first two principal components, with each dot representing an individual sample, color-coded by age in years. Right panel: Distribution of the first principal component (PC1) across different age groups shown as boxplots with overlaid jitter plots. The boxes represent the interquartile range (IQR; 25th to 75th percentiles), with the median value indicated by a central line. Whiskers extend to the minimum and maximum values within 1.5 times the IQR from the box. A legend on the right specifies the number of samples per age group. (C) Left panel: Hierarchical clustering analysis of sex-specific proteins significantly associated with age in the urine proteome. The heat map displays z-scored mean protein intensities across sex and age. Right panel: Protein intensity trajectories across age of the five selected modules from the left panel. Solid lines represent the mean values across age groups, and shaded areas indicate the 95% confidence intervals. (D) Same analysis as in (C), but for neat plasma samples.

meaningful low-abundance proteins. The opposite trends of modules 4 and 3 suggest the transition from innate immunity in young children to adaptive immunity as they grow older (Pieren et al, 2022). For Modules 1 and 2, we observe nonlinear trajectories: Module 1 shows a decrease in protein intensity until about age 10, then levels off until age 17. Conversely, Module 2 remains stable until about age 12, then shows a decrease, with females experiencing a sharper decline. Lastly, Module 5 shows a linear increase in protein intensities across age for males, while in females, protein intensities rise until age 12 and then begin to gradually decrease.

Our findings corroborate and expand on previous studies, confirming that both biological sex and age significantly contribute to interindividual variability in body fluids. These results highlight the importance of accounting for age and sex in clinical studies of pediatric disorders.

## Detection of pediatric disease-related proteome changes in urine and plasma

Our cohort includes 394 distinct pediatric diagnoses, representing a broad spectrum of clinical and biological variability. As illustrated in Fig. 1B, the ten most prevalent conditions in our dataset are cystic fibrosis ($n = 79$), celiac disease ($n = 68$), diabetes mellitus type 1 ($n = 29$), Crohn's disease ($n = 25$), adrenogenital disorder ($n = 22$), hemophilia A ($n = 19$), short stature disorder ($n = 18$), disorder of immune function ($n = 17$), chronic ulcerative colitis ($n = 16$), and familial Mediterranean fever ($n = 15$). For these highly represented disease groups, we performed analysis of covariance (ANCOVA) with sex and age as covariates, using the control group ($n = 131$) as the reference. This analysis revealed significant differential protein expression in the diseases relative to the healthy controls.

In cystic fibrosis, the most prevalent condition, we identified 78 significantly regulated protein groups in the neat plasma proteome (Fig. 3A). Among these, aldolase B (ALDOB) and cadherin 1 (CDH1) were the most significantly upregulated (lowest FDR-adjusted $P$ values), while immunoglobulin delta heavy chain (IGHD) exhibited the largest downregulation. Cystic fibrosis (CF) is one of the most common autosomal recessive disorders in Caucasian populations, with one in 25-30 individuals being carriers of the mutations in the cystic fibrosis transmembrane conductor regulatory (CFTR) gene (Sanders and Fink, 2016). This protein is critical for maintaining fluid homeostasis and regulating epithelial ion and water transport. Loss of CFTR function leads to the accumulation of thick mucus in the respiratory (Mall et al, 2024) and gastrointestinal tracts (Ooi and Durie, 2016), contributing to chronic inflammation and organ dysfunction. In our study, ALDOB showed elevated expression in CF patients compared not only to healthy controls but also to individuals with the ten most prevalent diseases in our dataset (Fig. 3B, top panel). ALDOB, a key glycolytic enzyme, is linked to metabolic reprogramming in the colonic mucosa (Bu et al, 2018). RNA-seq data from CF patients' colonic tissue suggest that increased ALDOB expression may be part of an adaptive response to metabolic stress in the gastrointestinal tract (Dayama et al, 2020). Similarly, CDH1 (cadherin 1), a calcium-dependent cell adhesion molecule, was significantly upregulated in CF compared to both healthy controls and the other prevalent diseases (Fig. 3B, middle panel). CDH1 plays a pivotal role in epithelial integrity and cell-cell adhesion (Van Roy and Berx,

2008). In the context of CF, dysregulation of CDH1 can profoundly impact pediatric lung development by altering epithelial-mesenchymal transition (EMT) processes, which are critical for tissue repair and remodeling. This dysregulation may contribute to impaired immune responses and exacerbate chronic airway inflammation, hallmark features of CF pathology. Elevated expression of CDH1 has been observed in CFTR knockout models of rat lung tissue, suggesting a potential mechanistic link between CFTR dysfunction and the dysregulation of EMT pathways (Rout-Pitt et al, 2024). Such aberrant epithelial responses likely contribute to pathological tissue remodeling, including the fibrosis and structural abnormalities characteristic of advanced CF. In contrast, IGHD, the delta heavy chain of immunoglobulin D (IgD), was markedly downregulated in CF compared to healthy controls (Fig. 3B, bottom panel). Secreted IgD plays a critical role in regulating mucosal homeostasis, particularly through its interaction with the commensal microbiota in mucosal tissues (Gutzeit et al, 2018). The observed reduction in IGHD aligns with impaired mucosal immune responses in CF and other diseases associated with mucosal barrier dysfunction, such as Crohn's disease and celiac disease.

To improve sensitivity for low-abundance disease-related markers, we complemented the neat plasma analysis with PCA-N plasma measured on the Thermo Orbitrap Astral MS. This revealed an additional 67 significantly differentially expressed protein groups between cystic fibrosis patients and healthy individuals (Fig. 3C), including GUCA2A, ADAMDEC1, and FABP1 (upregulated) and GP2 (downregulated) (Fig. 3D). These proteins highlight key pathophysiological processes in CF, such as a compensatory response by guanylin (GUCA2A) to restore proper fluid balance in the intestine (Sindic, 2013), increased chronic inflammation and extracellular matrix remodeling by ADAM-like decysin 1 (ADAMDEC1) (Jasso et al, 2022), and compensatory response by fatty acid-binding protein 1 (FABP1) to altered lipid metabolism in the liver (Peretti et al, 2005). Glycoprotein 2 (GP2) in pancreatic cells is involved in immune surveillance of the gut and its downregulation in CF may reflect pancreatic dysfunction and impaired mucosal immunity (Zhang et al, 2024).

In contrast, no significantly regulated proteins were observed in the urine proteome of CF patients. Although urinary incontinence is common among female adolescents with CF due to chronic coughing and pelvic floor dysfunction (Nankivell et al, 2010), its impact on the proteome is likely indirect, possibly contributing to interindividual variability through urothelial protein shedding or dilution effects from increased fluid intake.

On the other hand, we observed two significantly upregulated proteins in the urine proteome of patients with adrenogenital disorder: haptoglobin (HP) and F-actin capping protein subunit alpha-2 (CAPZA2) (Fig. 3E). Adrenogenital disorder or chronic adrenal hyperplasia primarily affects steroid hormone synthesis, which has an underlying overproduction of reactive oxygen species (ROS) that could potentially result in oxidative stress (Prasad et al, 2014; Bertaggia et al, 2014). HP is an acute-phase protein involved in oxidative stress regulation, and its upregulation (Fig. 3F, top panel) may reflect systemic inflammation or increased oxidative burden associated with adrenal steroid hormone dysregulation. CAPZA2, which regulates actin filament dynamics (Huang et al, 2020), has not been previously associated with adrenogenital disorder. Its upregulation (Fig. 3F, bottom panel) may suggest androgen-induced cytoskeletal remodeling in renal or urothelial cells.

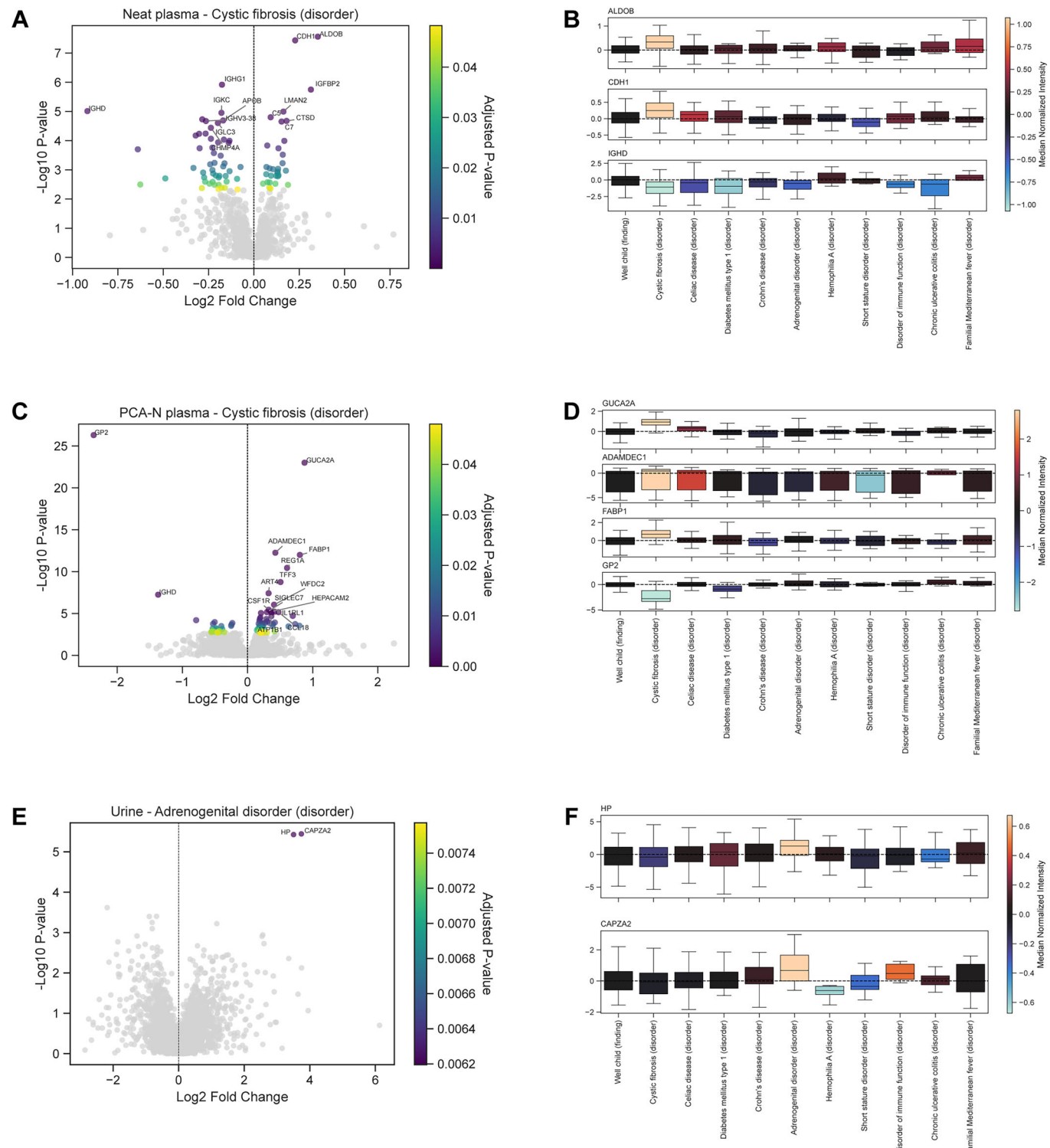

To widen the scope of our analysis, we systematically assessed the proteomic impact of the ten most prevalent diseases in our dataset by performing differential expression analysis in both urine and plasma proteomes relative to healthy controls (Fig. EV4; Dataset EV3). While not all conditions exhibited statistically significant hits after FDR correction, this analysis provides valuable insight into disease-specific proteomic variability and underscores how sample size influences statistical outcomes. These results highlight the utility of our dataset as a resource for designing future studies, particularly for estimating variance and statistical power requirements for different cohort sizes and study designs. Building on these insights from the most prevalent diseases, we

**Figure 3.  Differential protein expression in the ten most prevalent pediatric diseases.**

Sample sizes: control group, $n = 131$; cystic fibrosis, $n = 79$; celiac disease, $n = 68$; diabetes mellitus type 1, $n = 29$; Crohn's disease, $n = 25$; adrenogenital disorder, $n = 22$; hemophilia A, $n = 19$; short stature disorder, $n = 18$; disorder of immune function, $n = 17$; chronic ulcerative colitis, $n = 16$; and familial Mediterranean fever, $n = 15$. (**A**) Volcano plot showing differential protein expression in the neat plasma proteome of cystic fibrosis patients ($n = 79$) compared to healthy controls ($n = 131$), analyzed using ANCOVA with sex and age as covariates. Proteins with significant changes (FDR-adjusted $P$ value $< 0.05$) are highlighted. This panel is also shown as part of the representative plots in Fig. EV4D. (**B**) Boxplots comparing aldolase B (ALDOB), cadherin 1 (CDH1), and immunoglobulin delta heavy chain (IGHD) expression in the neat plasma proteome of the control group labeled as "Well child (finding)" and the ten most prevalent diseases in the cohort. Protein intensities are normalized against the median protein intensity of the control group. The boxes represent the interquartile range (IQR), with the median indicated by a central line. Whiskers extend to the farthest point within 1.5 times the IQR from the box. Outliers are removed for clarity. A horizontal dashed line marks the zero point (median protein intensity of the control group). (**C**) Same as in (**A**) but for the PCA-N plasma proteome. This panel is also shown as part of the representative plots in Fig. EV4E. (**D**) Same as in (**B**) but for the expression of guanylin (GUCA2A), ADAM-like decysin 1 (ADAMDEC1), fatty acid-binding protein 1 (FABP1), and glycoprotein 2 (GP2) in the PCA-N plasma proteome. Protein intensities are normalized against the median protein intensity of the control group. The boxes represent the interquartile range (IQR), with the median indicated by a central line. Whiskers extend to the farthest point within 1.5 times the IQR from the box. Outliers are removed for clarity. A horizontal dashed line marks the zero point (median protein intensity of the control group). (**E**) Volcano plot showing differential protein expression in the urine proteome of adrenogenital disorder patients ($n = 22$) compared to healthy controls ($n = 131$), analyzed using ANCOVA with sex and age as covariates. Proteins with significant changes (FDR-adjusted $P$ value $< 0.05$) are highlighted. This panel is also shown as part of the representative plots in Fig. EV4C. (**F**) Boxplots comparing haptoglobin (HP) and F-actin-capping protein subunit alpha-2 (CAPZA2) expression in the urine proteome of the control group labeled as "Well child (finding)" and the ten most prevalent diseases in the cohort. Protein intensities are normalized against the median protein intensity of the control group. The boxes represent the interquartile range (IQR), with the median indicated by a central line. Whiskers extend to the farthest point within 1.5 times the IQR from the box. Outliers are removed for clarity. A horizontal dashed line marks the zero point (median protein intensity of the control group).

next developed strategies for analyzing rare diseases within the cohort.

## Integration of SNOMED CT ontology to proteomics

Analyzing rare diseases in our pediatric cohort poses unique challenges, as 344 disorders in our dataset are represented by fewer than five patients each (Fig. EV5A, left panel). These small sample sizes hinder statistical analysis and limit the ability to perform robust proteomic characterization. To address this limitation, we leveraged the Systematized Nomenclature of Medicine Clinical Terms (SNOMED CT) ontology to create a framework that integrates these rare disorders into broader, biologically meaningful groupings. SNOMED CT provides a hierarchical structure of medical terms, enabling the systematic analysis of relationships among conditions.

To enable clustering of rare pediatric disorders despite small patient numbers, we constructed a cohort-specific disease ontology based on SNOMED CT and embedded it into a low-dimensional latent space using node2vec (Fig. EV5B). We empirically tested different depths of ancestry and descent to improve graph connectivity and found that including up to two levels provided optimal hierarchical context while avoiding overly fragmented subgraphs (Fig. EV5C). We then applied unsupervised clustering to group diagnoses by semantic similarity, systematically optimizing parameters based on Silhouette scores (Fig. EV5D–F). This approach reduced the number of underpowered disease categories (<5 patients) from 344 to 8, enabling statistical comparisons across aggregated groups (Fig. EV5A, right panel). The resulting clusters were evaluated for biological coherence using both expert curation (Dataset EV4) and pairwise embedding distances (Fig. 4A). A UMAP projection of the disease embeddings (Fig. 4B) further illustrated clear cluster separation, with examples such as Cluster 5 (infectious diseases) and Cluster 12 (immune-mediated disorders) demonstrating biologically meaningful groupings. Full details of the ontology graph construction, embedding, and clustering procedure are provided in "Methods".

We then integrated the clusters derived from semantic relationships and network embeddings with our proteomics data. As an initial step, we calculated the interindividual coefficients of variation (CVs) for protein intensities within each cluster. In the urine proteome, Cluster 18 exhibited the lowest interindividual CV (Fig. 4C, left panel), indicating high proteomic consistency across its samples. This cluster includes structural heart conditions such as Congestive cardiomyopathy, Hypoplastic left heart syndrome, Long QT syndrome, and Transplanted heart present. The bottom panel of Fig. 4C shows a tree diagram connecting these diseases to their most distant shared parent term: Heart disease (disorder). The right panel displays the overall biological CV across all quantified proteins in the urine proteome (with outliers removed from both panels). The overall biological CV was 55.9%, substantially higher than the CV of any individual cluster. This difference suggests that the clustering method effectively identifies patient subgroups with more consistent proteomic patterns, leading to reduced variability within clusters relative to the overall dataset.

Meanwhile, in the PCA-N plasma proteome, Cluster 16 showed the lowest interindividual CV (Fig. 4D, left panel). This cluster consists of musculoskeletal disorders affecting the knee, including Arthropathy of the knee joint and Juvenile osteochondrosis of the tibial tubercle. The bottom panel of Fig. 4D shows that both of these diseases connect to the shared distant parent term: Disease of lower extremity (disorder). The right panel presents the overall biological CV in the PCA-N plasma proteome, which was 47.9%. As in the urine proteome, the overall CV was higher than that of any cluster, reinforcing that semantic embedding-based clustering reduces variability and captures biologically coherent patient subgroups.

Building on these observations, we next tested whether these low-CV clusters exhibited distinct proteomic signatures compared to healthy controls. To address unequal variance and unbalanced sample sizes between the groups, we applied Welch's $t$ test for differential expression analysis. In Cluster 18 of the urine proteome dataset, this approach identified 66 protein groups with significantly altered expression relative to controls (Fig. 4E), using an

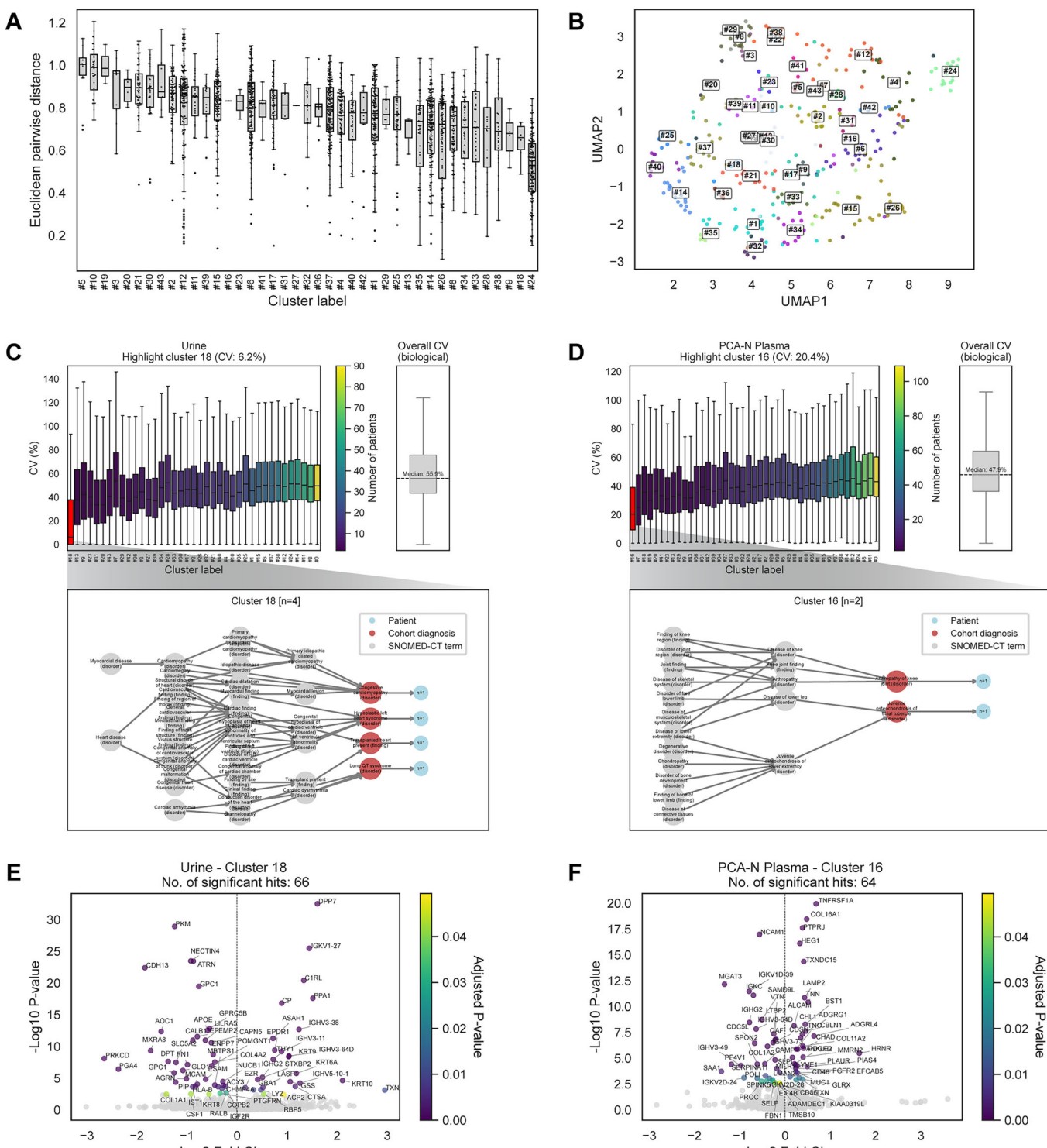

FDR-adjusted *P* value threshold (Benjamini–Hochberg correction) of <0.05. Among the upregulated proteins, ceruloplasmin (CP), an independent predictor of long-term all-cause mortality in patients with heart failure (Hammadah et al, 2014), and thioredoxin (TXN), a redox regulator previously linked to chronic heart failure (Jekell et al, 2004; Sánchez-Villamil et al, 2016), are particularly notable. Conversely, protein kinase C delta (PRKCD) was significantly

downregulated. Given PRKCD's role in cardiomyocyte apoptosis and oxidative stress signaling and its association with adverse cardiac remodeling (Miao et al, 2022), its downregulation in this cluster is unexpected. It may reflect distinct regulatory dynamics in pediatric structural heart disease, warranting further investigation.

In the PCA-N plasma proteome, Cluster 16 showed the lowest interindividual coefficient of variation (CV) (Fig. 4F). Differential

**Figure 4. Analysis of SNOMED CT-based disease clusters and their proteome profiles.**

(A) Pairwise Euclidean distances between disease nodes within each cluster, calculated from node2vec embeddings. Lower distances indicate greater biological and clinical similarity. (B) Two-dimensional UMAP visualization of node2vec embeddings for SNOMED CT diagnosis nodes in the cohort. Each point represents a disease term, colored and labeled by cluster assignments from $k$-means clustering ($k = 43$). (C) Coefficient of variation (CV) analysis for urine proteome clusters. The left panel shows boxplots representing the interindividual CVs for protein intensities within each cluster. Cluster 18 ($n = 4$), highlighted in red, exhibits the lowest CV among the clusters. The right panel shows the overall biological CV across all quantified proteins in the urine proteome. In both panels, boxplots show the distribution of values across groups: the center line indicates the median, box limits represent the interquartile range (25th to 75th percentiles), and whiskers extend to the most extreme data point within 1.5 times the IQR. Outliers were removed for visualization clarity. The accompanying tree diagram below illustrates the SNOMED CT terms and the number of patients associated with cluster 18. (D) Coefficient of variation (CV) analysis for PCA-N plasma proteome clusters. The left panel shows boxplots representing the interindividual CVs for protein intensities within each cluster. Cluster 16 ($n = 2$), highlighted in red, exhibits the lowest CV among the clusters. The right panel shows the overall biological CV across all quantified proteins in the PCA-N plasma proteome. In both panels, boxplots show the distribution of values across groups: the center line indicates the median, box limits represent the interquartile range (25th to 75th percentiles), and whiskers extend to the most extreme data point within 1.5 times the IQR. Outliers were removed for visualization clarity. The accompanying tree diagram below illustrates the SNOMED CT terms and the number of patients associated with cluster 16. (E) Volcano plot of differential protein expression in the urine proteome between cluster 18 ($n = 4$) and healthy controls ($n = 131$), using Welch's $t$ test with FDR-adjusted $P$ value < 0.05. Proteins with significant expression differences are highlighted and color-coded based on their adjusted $P$ values. (F) Volcano plot of differential protein expression in the PCA-N plasma proteome between cluster 16 ($n = 2$) and healthy controls ($n = 131$), using Welch's $t$ test with FDR-adjusted $P$ value < 0.05. Proteins with significant expression differences are highlighted and color-coded based on their adjusted $p$ values.

expression analysis using Welch's $t$ test identified 64 protein groups with significantly altered expression compared to healthy controls (Fig. 4F), using an FDR-adjusted $P$ value threshold (Benjamini–Hochberg correction) of <0.05. Among the upregulated proteins were collagen type XVI alpha-1 (COL16A1), a cartilage extracellular matrix component implicated in connective tissue integrity (Grässel and Bauer, 2013), and tenascin-N (TNN), a matricellular protein involved in tissue remodeling (Jones and Jones, 2000). Notably, neural cell adhesion molecule 1 (NCAM1) was significantly downregulated. NCAM1 plays a key role in myoblast fusion and musculoskeletal development (Knudsen et al, 1990), and its reduced expression may reflect impaired regenerative processes at the muscle–bone interface relevant to the knee joint disorders in this cluster.

These two cluster examples (Clusters 18 and 16) underscore the key advantage of our ontology-guided clustering approach in overcoming the limitations of small sample sizes. For instance, in Cluster 18 of the urine proteome, each of the four disease nodes was represented by only a single patient in our cohort (Fig. 4C,E). Individually, these rare conditions would not permit differential expression analysis due to the lack of within-group variance. However, by grouping them based on shared semantic and clinical relationships within the SNOMED CT framework, we aggregated a sufficient sample size to identify 66 significantly regulated proteins relative to controls. Similarly, Cluster 16 in the PCA-N plasma proteome grouped together two rare musculoskeletal conditions with shared clinical characteristics, each represented by a single patient in our cohort (Fig. 4D,F). This resulted in a low-CV cluster that also revealed robust proteomic differences. These examples demonstrate how clustering enhances statistical power while preserving biological relevance, providing a tractable solution for studying rare diseases in a unified framework.

Using this clustering framework, we identified a total of 838 significantly altered protein groups in urine, 393 in neat plasma, and 709 in PCA-N plasma (Fig. EV5G). Clusters with no significant hits in a given body fluid may reflect either the absence of patient samples for that fluid or the lack of statistically significant differences compared to healthy controls. Notably, the number of significant proteins detected in PCA-N plasma was nearly double that of the neat plasma dataset, further underscoring the value of

the perchloric acid-based depletion strategy in enhancing proteomic depth and sensitivity.

## Discussion

Understanding rare pediatric disorders has traditionally been hampered by small patient numbers, making it difficult to draw statistically meaningful conclusions about individual conditions. Our study demonstrates how this limitation can be addressed by integrating network-based clinical ontologies with molecular profiling data, enabling the analysis of rare diseases in aggregate rather than in isolation. By profiling 1140 children and adolescents representing 394 distinct disorders, we demonstrate the power of this framework in extracting meaningful insights from heterogeneous patient populations.

Through comprehensive proteomic analysis of both urine and plasma samples, we uncovered distinct molecular signatures reflecting tissue-specific functions and disease processes. By the identification of over 5000 proteins in urine, 900 in undepleted (neat) plasma, and 1900 in depleted (PCA-N) plasma, we revealed fluid-specific protein patterns. For instance, plasma profiles highlighted systemic processes through apolipoproteins and coagulation factors, while urine profiles reflected local tissue dynamics through epithelial and structural proteins. This dual-fluid approach provided complementary insights into both systemic and organ-specific disease manifestations.

Our data revealed distinct patterns in how biological and clinical variables affect urine and plasma proteomes. Specifically, we identified numerous differentially regulated proteins in urine when performing ANOVA across different age groups and sexes, whereas urine proteomes showed relatively few significant changes in response to disease status (healthy vs. diagnosed). In contrast, plasma proteomes exhibited the opposite trend, where disease status strongly influenced protein expression, with many differentially regulated proteins identified, whereas age and sex contributed far fewer significant changes.

These differences could be attributed to the distinct origins and roles of the two biofluids. Urine reflects localized processes occurring within the urinary tract and kidneys, where protein expression is likely influenced by developmental changes, such as

renal maturation and puberty, as well as by sex-specific physiology, including hormonal regulation. These factors can shape the urinary proteome indirectly through their impact on renal function and protein excretion. In addition, urine is uniquely sensitive to changes in the early stages of disease, particularly in renal and urinary tract conditions, due to its limited systemic buffering (Zhang et al, 2022). However, this sensitivity is disease- and tissue-specific. In systemic conditions such as cardiovascular disease, plasma biomarkers such as hsCRP, interleukins, and troponins remain the gold standard for early detection, owing to their direct release into circulation (Ridker et al, 2004; Kukova et al, 2019). These distinctions highlight the complementary roles of plasma and urine: while plasma reflects systemic inflammation and immune responses, urine captures tissue-localized changes that may otherwise be missed. Analyzing both fluids in parallel provides a more comprehensive understanding of disease biology.

The power of our integrated approach is particularly evident in how it handles rare conditions. By using network-based clustering of SNOMED CT terms and concepts, we could analyze conditions with only a single patient by connecting them to biologically related disorders, thereby gaining statistical power while maintaining clinical relevance. For example, our analysis of the urine proteome of patients with structural heart conditions revealed specific protein signatures associated with oxidative stress and cardiac pathology, insights that would have been statistically impossible to obtain from individual rare conditions alone. Similarly, our analysis of the depleted (PCA-N) plasma proteome of patients with musculoskeletal disorders of the knee revealed upregulation of proteins involved in extracellular matrix structure and tissue remodeling. These examples demonstrate how ontology-based clustering enables robust statistical comparisons even for diseases that are underpowered when considered individually.

A key factor contributing to the success of this ontology-based clustering approach lies in the depth of our proteomic data. Through the implementation of a perchloric acid-based depletion strategy, we more than doubled the number of protein group identifications in plasma and enriched for low-abundance proteins, as expected (Albrecht et al, 2025). This expanded proteomic coverage provided the resolution necessary to detect meaningful molecular signatures within the aggregated clusters, allowing us to uncover shared biological features across grouped rare diseases. These findings highlight the synergistic potential of combining deep molecular data from proteomics with clinically informed ontological structures in discovery-driven research.

Our framework also provides a bridge between clinical practice and molecular research by utilizing the same ontological structure (SNOMED CT) that underlies many electronic health record systems. This alignment could facilitate the translation of molecular findings into clinical applications and enable more systematic integration of molecular data into routine clinical care. The approach could be extended beyond proteomics to other molecular profiling techniques, such as metabolomics, transcriptomics, or multi-omics integration, potentially revealing additional layers of biological insight. However, several limitations of our approach to integration with any omics should be considered for future improvements. The quality and comprehensiveness of the underlying clinical ontology significantly impact the effectiveness of disease clustering. While SNOMED CT is widely used, it may not capture all relevant biological relationships between conditions, and some rare diseases may be inadequately represented.

As a discovery-based, proof-of-concept study, our initial aim was to explore the possible impact of proteomic profiling for underpowered pediatric patient groups. By ontology-guided clustering, we were able to uncover coherent and biologically meaningful proteomic patterns, and in this context, the recapitulation of known biology serves as an important validation of the approach. While these findings align with established knowledge, they demonstrate that the framework can recover relevant biology from heterogeneous clinical data. Future studies will be needed to extend this strategy toward clinical translation for prospective validation of any new potential markers we uncover in this study.

In conclusion, while our study focused on pediatric disorders, the framework we developed represents a generalizable approach for studying heterogeneous patient populations. By combining network-based clustering of clinical ontologies and molecular profiling, we create a path forward for understanding rare diseases not as isolated entities, but as part of larger disease networks with shared biological mechanisms.

# Methods

**Reagents and tools table**

| Reagent/resource | Reference/source | Identifier/catalog No. |
| --- | --- | --- |
| **Chemicals, enzymes, and other reagents** | | |
| Ammonium acetate | Merck | 1.01116 |
| Chloroacetamide (CAA) | Sigma-Aldrich | C0267 |
| n-Dodecyl β-D-Maltoside (DDM) | Sigma-Aldrich | D4641 |
| Dithiothreitol (DTT) | Sigma-Aldrich | D0632 |
| Formaldehyde solution (36.5-38% in $H_2O$): 'light' ($CH_2O$) | Sigma-Aldrich | F8775 |
| Formaldehyde-$^{13}$C,d$_2$-solution (20 wt. % in $D_2O$): 'heavy' ($^{13}CD_2O$) | Sigma-Aldrich | 596388 |
| Formaldehyde-d$_2$-solution (20 wt. % in $D_2O$): 'intermediate' ($CD_2O$) | Sigma-Aldrich | 492620 |
| Optima® LC/MS-grade Formic acid (FA) | Fisher Chemical | A117-50 |
| Optima® LC/MS-grade Acetonitrile (ACN) | Fisher Chemical | A955-212 |
| Optima® LC/MS-grade water | Fisher Chemical | W6-4 |
| Lichropur® ammonia solution, 25% | Merck | 5.33003 |
| Lysyl Endopeptidase, Mass spectrometry Grade (LysC) | FUJIFILM Wako | 125-05061 |

| Reagent/resource | Reference/source | Identifier/catalog No. |
|---|---|---|
| MagReSyn® HILIC beads, 20 mg/mL suspension in 20% EtOH | ReSyn Biosciences | MR-HLC010 |
| Sodium cyanoborodeuteride: 'heavy' (NaBD$_3$CN) | Sigma-Aldrich | 190020 |
| Sodium cyanoborohydride: 'light' (NaBH$_3$CN) | Sigma-Aldrich | 156159 |
| Sodium dodecyl sulfate (SDS) | Carl Roth | CN30.3 |
| Triethylammonium bicarbonate (TEAB) buffer, 1.0 M, pH 8.5 ± 0.1 | Sigma-Aldrich | T7408 |
| Tris(2-carboxyethyl) phosphine hydrochloride (TCEP-HCl) | ThermoFisher Scientific | 20491 |
| Trizma® pre-set crystals, pH 8.5 | Sigma-Aldrich | T8818 |
| Trypsin, proteomics grade | Sigma-Aldrich | T6567 |
| Urea | Sigma-Aldrich | U1250 |
| **Software** | | |
| MaxQuant | https://maxquant.net | version 2.0.1.0 |
| DIA-NN | https://github.com/vdemichev/DiaNN | version 1.8.1 and version 1.8.2 beta 34 |
| DirectLFQ | https://github.com/MannLabs/directlfq | version 0.2.5 |
| RefQuant | https://github.com/MannLabs/refquant | N/A |
| Perseus | https://maxquant.net/perseus | version 2.0.11 |
| Jupyter Notebook | https://jupyter.org | version 7.1.2 |
| Python | https://python.org | Version 3.12.2 |
| Breaking cycles in noisy hierarchies | https://github.com/zhenv5/breaking_cycles_in_noisy_hierarchies | N/A |
| **Others** | | |
| twin.tec® PCR Plate 96-well LoBind | Eppendorf | 0030129512 |
| twin.tec® PCR Plate 384-well LoBind | Eppendorf | 0030129547 |
| Deepwell Plate 96-well, 1000 μL | Eppendorf | 951032603 |
| ThermoMixer® C | Eppendorf | 5382000015 |
| Mastercycler™ X50h | Eppendorf | 6316000019 |
| Heat Sealer S200 | Eppendorf | 5392000030 |
| DynaMag™-2 Magnet | ThermoFisher Scientific | 12321D |
| DynaMag™-96 Side Skirted Magnet | ThermoFisher Scientific | 12027 |

| Reagent/resource | Reference/source | Identifier/catalog No. |
|---|---|---|
| Deep Well MagnaBot® 96 Magnetic Separation Device | Promega | V3031 |
| Nanodrop 2000 Spectrophotometer | ThermoFisher Scientific | ND-2000 |
| Infinite® M Plex multimode microplate reader | Tecan | 30190085 |
| Bravo robot | Agilent | N/A |

## Methods and protocols

### Study cohort

In this study, urine ($n = 1000$) and plasma ($n = 999$) samples were analyzed from a pediatric cohort consisting of 1140 participants, including 131 healthy individuals and 1009 diagnosed with various pediatric disorders. The participants were recruited at the Dr. von Hauner Children's Hospital in Munich between January 2021 and March 2023. All samples were collected with informed consent and in compliance with the hospital's approved protocols (Nr. 20-172), adhering to the Declaration of Helsinki. In addition, all procedures involving human participants conformed to the ethical principles set out in the Belmont Report issued by the Department of Health and Human Services.

Urine was collected by self-sampling at patient visit in the clinic. Upon collection, 1.5 mL of urine was aliquoted into a 2 mL Sarstedt tube, centrifuged at $2000 \times g$ and 4 °C for 5 min, and stored at −80 °C.

For plasma collection, blood was drawn during routine medical checks. Briefly, blood was collected in EDTA tubes to prevent coagulation and centrifuged at $2000 \times g$ for 15 min to separate cellular components from the plasma. The plasma was then aliquoted and stored at −80 °C. Complete blood counts (CBC) and comprehensive metabolic panels (CMP) were routinely obtained from each patient as part of the diagnostic process. If changes in blood counts suggested a potential infectious disease, additional tests were conducted for further assessment.

### Clinical data extraction and management

Clinical and demographic data—including age, sex, height, weight, vital signs, imaging results, and diagnoses—were collected for each participant from the hospital's electronic health record system (KAS) and transferred into the CentraXX software platform, which served as the central repository for clinical study data. All relevant patient data were extracted from CentraXX and manually curated by trained researchers to ensure completeness and accuracy. Diagnoses were reviewed and verified by the treating pediatric specialists and encoded using standardized medical terminologies (ICD-10 and SNOMED CT), with Orpha codes applied for rare conditions. Medications were recorded according to the Anatomical Therapeutic Chemical (ATC) classification. To enable structured representation of clinical features, phenotypic abnormalities were annotated using the Human Phenotype Ontology (HPO). Additional clinical information not captured in the EHR was collected via structured electronic questionnaires (LimeSurvey).

All clinical data were pseudonymized and stored securely at the hospital, accessible only to authorized study personnel in compliance with data protection regulations.

### Sample preparation

**Urine samples for quality assessment (reference channel)**: To prepare the reference channel for the mDIA workflow of our urine proteomics dataset, we pooled 20 μL from each urine sample across the cohort ($n = 1000$), resulting in a total pooled volume of 20 mL. We employed a modified urine-HILIC protocol based on an established on-bead protein capture, cleanup, and digestion method (Govender et al, 2023). Briefly, we added three times the volume (60 mL) of lysis buffer (8 M urea, 2% SDS in 50 mM Tris, pH 8.0) to the pooled urine, followed by the addition of 2 mL of 410 mM dithiothreitol (410 mM DTT in 50 mM Tris, pH 8.0) to achieve a final DTT concentration of 10 mM. The mixture was then incubated for 30 min at room temperature. Subsequently, reduced cysteine residues were alkylated using 2.6 mL of 960 mM chloroacetamide (960 mM CAA in 50 mM Tris, pH 8.0), achieving a final concentration of 30 mM CAA. The mixture was again incubated in the dark for 30 min at room temperature.

During incubation, we separately prepared the MagReSyn® HILIC beads (ReSyn Biosciences) by placing 2 mL of a 20 μg/μL bead suspension (1:10 v/v beads-to-sample ratio) on a magnetic separator, removing the shipping solution, and washing the beads two times with equilibration buffer (15% acetonitrile [ACN] in 100 mM ammonium acetate, pH 4.5). The beads were finally equilibrated in an equal volume (2 mL) of the equilibration buffer.

After lysis, reduction, and alkylation of the urine sample, we combined the sample lysate with an equal volume (84.6 mL) of binding buffer (30% ACN in 200 mM ammonium acetate, pH 4.5) and added 2 mL of the equilibrated beads. We continuously and gently mixed the bead-sample mixture for 30 min at room temperature to ensure adequate interaction. After allowing the beads to settle, the supernatant was decanted until a sufficient volume remained to facilitate separation using a magnetic separator. The beads, with bound proteins, were washed twice with 95% ACN, discarding the wash each time. Finally, the beads were resuspended in 5 mL of digestion buffer (100 mM TEAB, pH 8.0). Given that urine samples are estimated to contain ~0.2 μg/μL of protein, we added 80 μg each of trypsin and LysC (1:50 w/w enzyme-protein ratio) for overnight digestion at 37 °C.

After digestion, the resulting peptide solution was removed from the beads using a magnetic separator and transferred into a clean tube. Peptide concentration was measured optically at 280 nm using a Nanodrop 2000 (Thermo Scientific). The peptides were then labeled immediately with the dimethyl light (Δ0) channel for mDIA measurement as described by Thielert et al, Briefly, for every 20 μg of peptides, 4 μL of freshly prepared 4% (v/v) formaldehyde ($CH_2O$) and 4 μL of freshly prepared 600 mM sodium cyanoborohydride ($NaBH_3CN$) were sequentially added to the sample solution. The mixture was incubated at room temperature for 1 h on a bench-top mixer. Subsequently, for every 20 μg of peptides, 16 μL of 1% (v/v) ammonia ($NH_4OH$) was added to quench the reaction. Finally, the solution was acidified to pH 2–3 using formic acid, and an 8 ng/μL working solution was prepared for all subsequent analyses.

**Urine samples for labeling efficiency and FDR assessment**: From the 8 ng/μL working solution of the Δ0-labeled urine sample,

six replicates, each containing 200 ng of peptides, were prepared. Three replicates were designated for assessing dimethyl labeling efficiency, and another three for assessing false discovery rate (FDR). The solutions were loaded onto Evotips (Evosep Biosystems) according to the manufacturer's instructions. Briefly, Evotips were activated with 1-propanol, washed with 0.1% formic acid in acetonitrile, equilibrated with 0.1% formic acid, loaded with the 600 ng samples, washed again with 0.1% formic acid, and maintained in 150 μL of 0.1% formic acid to prevent drying prior to analysis on the timsTOF HT.

**Urine samples for discovery proteomics (target channels)**: Urine samples for the target channels were prepared similarly to the pooled reference channel, except that they were processed individually in separate 96-well plates. Briefly, 100 μL of each urine sample was aliquoted using the Bravo robot into 96-deepwell plates containing 300 μL of lysis buffer (8 M urea, 2% SDS in 50 mM Tris, pH 8.0) in each well. Then, 10 μL of 410 mM dithiothreitol (410 mM DTT in 50 mM Tris, pH 8.0) was mixed into each well, and the plates were incubated for 30 min at room temperature. Subsequently, 13 μL of 960 mM chloroacetamide (960 mM CAA in 50 mM Tris, pH 8.0) was added, and the plates were incubated in the dark for an additional 30 min at room temperature.

During incubation, the MagReSyn® HILIC beads (ReSyn Biosciences) were prepared separately by aliquoting 12 mL of a 20 μg/μL bead suspension into twelve 1.5-mL Eppendorf tubes. The shipping solution was removed using a magnetic rack, and the beads were washed twice with equilibration buffer (15% ACN in 100 mM ammonium acetate, pH 4.5). The beads were then resuspended in an equal volume (1 mL) of equilibration buffer and pooled in a 15-mL conical tube. Finally, 10 μL of this bead suspension (1:10 v/v beads-to-sample ratio) was aliquoted into separate wells of 12 × 96-deepwell plates, with 25 μL of equilibration buffer added to each well to prevent the beads from drying out.

Using the Bravo robot, an equal volume (423 μL) of the binding buffer (30% ACN in 200 mM ammonium acetate, pH 4.5) was mixed with the sample lysate, and the entire mixture was transferred into the deepwell plates containing the beads. The sample was allowed to interact with the beads by incubating the mixture for 30 min at room temperature. After incubation, the supernatant was removed using a magnetic rack, the beads were resuspended in 150 μL of 95% ACN, and the suspension was transferred to a 96-well twin.tec® plates. The adsorbed proteins on the beads were washed twice more with 150 μL of 95% ACN. Finally, 100 μL of digestion buffer (100 mM TEAB, pH 8.0) containing 0.4 μg each of trypsin and LysC (1:50 w/w enzyme-protein ratio) was added to the beads, and they were incubated at 37 °C overnight for digestion.

After digestion, the resulting peptide solution was removed from the beads using a magnetic rack and transferred into clean, separate 96-well twin.tec® plates. The peptide concentration was measured using a tryptophan assay and normalized to 100 ng for each sample. The peptides were then immediately labeled with dimethyl medium (Δ4) and dimethyl heavy (Δ8) channels according to the following scheme: odd-numbered plates (plates 1, 3, 5, 7, 9, 11) had samples in rows A-D labeled with medium (Δ4) channels and rows E-H labeled with heavy (Δ8) channels, whereas even-numbered plates (plates 2, 4, 6, 8, 10, 12) had samples in rows A-D labeled with heavy (Δ8) channels and rows E-H labeled with medium (Δ4) channels. The labeling was performed as described above. For the

medium (Δ4) channel, formaldehyde-d2-solution (CD$_2$O) and sodium cyanoborohydride (NaBH$_3$CN) were used, while for the heavy (Δ8) channel, formaldehyde-$^{13}$C,d$_2$-solution ($^{13}$CD$_2$O) and sodium cyanoborodeuteride (NaBD$_3$CN) were sequentially added to the sample solution. After incubating for 1 h and quenching the reaction, the solution was acidified to pH 2–3 using formic acid.

Following the labeling process, the Δ4 and Δ8 channels were systematically combined. Specifically, peptides from odd-numbered plates with rows A-D labeled with Δ4 were mixed with those from corresponding even-numbered plates where rows A-D were labeled with Δ8. Conversely, for rows E-H, peptides labeled with Δ8 from odd-numbered plates were combined with peptides labeled with Δ4 from even-numbered plates. This pairing approach—plates 1 with 2, 3 with 4, continuing up to 11 with 12—allowed the integration of both labeling variants into each sample. In addition, 50 μL of the 8 ng/μL Δ0-labeled reference channel was added to each well, allowing the multiplexing of all three dimethyl channels (400 ng Δ0/100 ng Δ4/100 ng Δ8) in each well. This resulted in a total injection amount of 600 ng.

Finally, the multiplexed samples were loaded onto Evotips (Evosep Biosystems) according to the manufacturer's instructions, as described previously.

**Plasma samples for quality assessment**: To prepare the QC samples for plasma, we pooled 10 μL of each plasma sample from plate number 6 into a separate 1.5-mL Eppendorf tube. Subsequently, we added 1 μL of this pooled plasma sample into the last two columns of the 384-well twin.tec® plates containing 14 μL of lysis buffer (10 mM TCEP, 40 mM CAA, and 0.02% DDM in 100 mM Tris, pH 8.5) using the Bravo robot. These samples were then processed together with the other individual plasma samples.

**Neat plasma samples for discovery proteomics**: Plasma samples were prepared similarly to the pooled reference channel but were processed individually in separate 384-well plates. Briefly, 1 μL of each plasma sample was aliquoted into 384-well plates containing 14 μL of lysis buffer (10 mM TCEP, 40 mM CAA, and 0.02% DDM in 100 mM Tris, pH 8.5) using the Bravo robot. The mixture was then incubated at 95 °C for 10 min in a thermocycler. Subsequently, 2 μL of digestion mix, containing 0.25 μg/μL trypsin and 0.25 μg/μL LysC, was added to each well. This corresponds to 0.5 μg of each enzyme per plasma sample, which is estimated to contain 50 μg/μL of protein (1:100 w/w enzyme-to-protein ratio). The proteins were digested overnight at 37 °C.

After digestion, 0.1% formic acid was added to achieve a final volume of 100 μL, and the solution was loaded onto Evotips (Evosep Biosystems) according to the manufacturer's instructions, as described previously.

**PCA-N plasma samples for discovery proteomics**: The PCA-N workflow was semiautomated using the Bravo robot. Five μL of plasma was diluted in 25 μL of water and then mixed with 25 μL of 1 M perchloric acid (PCA) to get a final concentration of 0.5 M. The mixture was incubated at 4 °C for 60 min in 96-well plates. The suspension was then centrifuged at 4000 × g for 20 min at 4 °C, and 24 μL of the supernatant was then transferred to a 384-well plate. The pH was adjusted to pH 8–8.5 using precisely titrated sodium hydroxide solution (NaOH, 1.4 M, 8 μL) to provide optimal conditions for enzymatic digestion. Subsequently, proteins were reduced, alkylated, and denatured at 95 °C for 10 min using dithiothreitol (DTT, C$_{final}$ = 10 mM) and CAA (C$_{final}$ = 40 mM) in a triethylammonium bicarbonate buffer (TEAB, pH 8–8.5,

C$_{final}$ = 60 mM). n-Dodecyl β-D-maltoside (DDM, C$_{final}$ = 0.01%) was used as detergent, ensuring compatibility with subsequent C18-based peptide desalting. Proteins were digested using trypsin/LysC (each 0.125 μg/μL, 0.8 μL), and digestion was stopped with TFA (C$_{final}$ = 0.5%).

After digestion, 200 ng of the digested peptides were loaded onto Evotips (Evosep Biosystems) according to the manufacturer's instructions, as described previously.

### Data acquisition in LC-MS/MS

**Chromatography and mass spectrometry setup:** All data acquisitions were performed using an Evosep One liquid chromatography system (Evosep Biosystems). The chromatographic gradients were generated by a mobile phase consisting of 0.1% formic acid in LC/MS-grade water (buffer A) and 0.1% formic acid in acetonitrile (buffer B). Three chromatography methods were employed, depending on the sample:

- Dimethyl-labeled urine samples were analyzed using the 30SPD (samples per day) method, featuring a 44 min active gradient on a PepSep column (15 cm length, 75 μm ID, 1.5 μm C18 beads; Bruker Daltonics) maintained at 50 °C.
- Label-free neat plasma samples were analyzed using the 60SPD (samples per day) method, with a 21 min active gradient on a PepSep column (8 cm length, 150 μm ID, 1.5 μm C18 beads; Bruker Daltonics) also maintained at 50 °C.
- Label-free PCA-N plasma samples were analyzed using the 100 SPD (samples per day) method, featuring an 11.5 min active ingredient on an Aurora Rapid column (8 cm, 150 μm ID, IonOpticks) also maintained at 50 °C.

Urine and neat plasma samples were measured with a timsTOF HT mass spectrometer (Bruker Daltonics). In both setups, the analytical column was connected to a fused silica emitter with a 10 μm ID (Bruker Daltonics). Meanwhile, PCA-N plasma samples were measured with an Orbitrap Astral mass spectrometer (ThermoFisher Scientific). The analytical column was interfaced with an EASY-Spray Source with a spray voltage of 1900 V, and the Astral mass spectrometer was interfaced with a FAIMS Pro device operated with a total carrier gas flow of 3.5 L/min and a compensation voltage of −40 V.

**dda-PASEF for assessment of labeling efficiency in urine:** The efficiency of dimethyl labeling was assessed using three technical replicates, with each replicate containing 200 ng of dimethyl Δ0-labeled urine peptides. The timsTOF HT was operated in dda-PASEF mode, covering a full scan range from 100 to 1700 $m/z$ within an ion mobility (IM) range of 0.7–1.3 Vs cm$^{-2}$. Four PASEF scans were acquired per topN acquisition cycle, wherein precursors exceeding an intensity threshold of 2500 arbitrary units were selected for fragmentation. Precursors reaching a target value of 20,000 arbitrary units were excluded from further selection for 0.4 min. Singly-charged precursors were also excluded based on their positioning within the polygon filter of the m/z-IM plane.

**dia-PASEF for assessment of false discovery rate (FDR) in urine:** The false discovery rate in the mDIA setup was empirically assessed using three technical replicates, each containing 200 ng of dimethyl Δ0-labeled urine peptides. The timsTOF HT was operated in dia-PASEF mode, covering a full scan range from 100 to 1700 $m/z$ within an ion mobility (IM) range of 0.7 to 1.3 Vs cm$^{-2}$. The method used for

the measurements was optimized with the Python tool py_diAID to ensure maximum precursor coverage of Δ0-labeled urine tryptic peptides (Skowronek et al, 2022). The optimal dia-PASEF method consisted of one MS1 scan followed by twenty dia-PASEF scans, with two IM ramps per dia-PASEF scan. The isolation windows of the dia-PASEF scans were of variable widths that cover a *m/z* range of 350–1200 (Table EV1). The accumulation and ramp time was specified at 100 ms, and the capillary voltage was set to 1900 V. The collision energy was configured to increase linearly from 20 eV at $1/K0 = 0.6$ Vs cm$^{-2}$ to 59 eV at $1/K0 = 1.6$ Vs cm$^{-2}$.

**dia-PASEF for discovery proteomics in urine**: For discovery proteomics in urine, 500 multiplexed samples (injection amount of 600 ng) were measured using the dia-PASEF mode, with the same method as that used for the FDR assessment.

**dia-PASEF for discovery proteomics in neat plasma**: For discovery proteomics in neat plasma, 999 label-free plasma samples and 112 label-free QC samples (injection amount of 500 ng) were measured using the dia-PASEF mode covering a full scan range from 100 to 1700 *m/z* within an ion mobility (IM) range of 0.7–1.3 Vs cm$^{-2}$. The method used for the measurements was optimized with the Python tool py_diAID to ensure maximum precursor coverage of label-free tryptic peptides (Skowronek et al, 2022). The optimal dia-PASEF method consisted of one MS1 scan followed by twelve dia-PASEF scans, with two IM ramps per dia-PASEF scan. The isolation windows of the dia-PASEF scans were of variable widths that cover a *m/z* range of 350–1200 (Table EV2). The accumulation and ramp time was specified at 100 ms, and the capillary voltage was set to 1750 V. The collision energy was configured to increase linearly from 20 eV at $1/K0 = 0.6$ Vs cm$^{-2}$ to 59 eV at $1/K0 = 1.6$ Vs cm$^{-2}$.

**dia-PASEF for discovery proteomics in PCA-N plasma**: For discovery proteomics in PCA-N plasma, 999 label-free samples and 112 label-free QC samples (injection amount of 200 ng) were measured in DIA mode. MS1 scans (380–980 *m/z*) were acquired at a resolution of 120,000 with an AGC target of 500% and a maximum injection time of 3 ms. MSMS scans were acquired in the Astral analyzer with 4 Th windows (scan range 150–2000 *m/z*), a maximum injection time of 7 ms, and an AGC target of 500%. HCD fragmentation was performed with 25% normalized collision energy.

### Raw data processing

**Labeling efficiency assessment with MaxQuant:** MaxQuant version 2.0.1.0 (Cox and Mann, 2008) was used to process the triplicate dda-PASEF raw files to calculate the labeling efficiency of the dimethyl light (Δ0) channel. The dimethyl Δ0 channel was first configured as a variable modification in MaxQuant for both the N-terminus (Var DimethNter0) and lysine residues (Var DimethLys0). During the peptide search, these two configurations, along with Oxidation (M) and Acetyl (Protein-N-term), were selected as variable modifications. In addition, Carbamidomethyl (C) was configured as a fixed modification, and a maximum of three modifications per peptide was allowed. The search was performed against the human SwissProt FASTA database (taxonomy ID 9606; 20,368 protein entries and 20,160 gene entries), including contaminants. The PSM and protein false discovery rates (FDR) were both set to 0.01, as determined by a decoy database generated by reversing the sequences in the FASTA. Enzyme

specificity was set to Trypsin/P, with the maximum number of missed cleavages limited to two. All other settings were maintained at default values.

**FDR assessment with DIA-NN:** DIA-NN version 1.8.2 beta 34 (Demichev et al, 2020) was used to search the triplicate dia-PASEF raw files using an AlphaPeptDeep-predicted spectral library that is trained for dimethyl-labeled tryptic HeLa data. We configured the following settings in the DIA-NN GUI: Protein inference was set to "Genes", Neural network classifier to "Single-pass mode", Quantification strategy to "QuantUMS (high precision)", Cross-run normalization to "RT-dependent", Library Generation to "IDs, RT and IM Profiling", and Speed and RAM usage to "Optimal results". Mass accuracy was set to 15 ppm, and MS1 accuracy was set to 10 ppm, as recommended for dia-PASEF runs. The scan window radius was set to 6. The settings "Use isotopologues", "MBR", "Heuristic protein inference", and "No shared spectra" were also enabled. We added more commands in the DIA-NN command line GUI: (1) {--fixed-mod Dimethyl, 28.0313, nK}, (2) {--channels Dimethyl, 0, nK, 0:0; Dimethyl, 4, nK, 4.0251:4.0251; Dimethyl, 8, nK, 8.0444:8.0444}, (3) {--channel-spec-norm}, (4) {--original-mods}, (5) {--peak-translation}, (6) {--report-lib-info}, and (7) {-mass-acc-quant 10.0}.

**Urine discovery cohort analysis with DIA-NN:** DIA-NN version 1.8.2 beta 34 was executed on a high-performance computing (HPC) cluster to perform parallel searches of the 553 mDIA raw files, using the same AlphaPeptDeep-predicted spectral library and DIA-NN GUI settings described above.

The resulting DIA-NN output table was processed using the 'RefQuant' package implemented in Python (https://github.com/MannLabs/refquant). Within this package, the filtering threshold for channel q-value was set to <0.2, maintaining a false discovery rate (FDR) of 1%. From the filtered precursors, RefQuant quantifies precursor intensities by leveraging the high ratio of the reference channel relative to the target channels. The package also seamlessly incorporates directLFQ (https://github.com/MannLabs/directlfq) (Ammar et al, 2023) for quantifying protein group intensities.

**Plasma (neat and PCA-N plasma) discovery cohort analysis with DIA-NN:** DIA-NN version 1.8.1 was executed on a high-performance computing (HPC) cluster to perform parallel searches of the 1111 raw files (999 in the discovery cohort + 112 pooled QC). The library-based search applied a spectral library previously generated from the human SwissProt FASTA database (taxonomy ID: 9606; 20,400 protein entries and 20,180 gene entries). This library was generated through in silico tryptic digestion, allowing a maximum of one missed cleavage. Modifications included cysteine carbamidomethylation as a fixed modification and both methionine oxidation and N-terminal acetylation as variable modifications, with only one variable modification permitted per peptide. Peptide lengths were set to range from 7 to 30 amino acids. The search parameters were configured with an MS1 mass accuracy of 10 ppm and MS2 mass accuracy of 15 ppm, and a scan window radius of 6. Match-between-runs was enabled to enhance identification consistency across samples. All other settings were maintained at default values. The resulting DIA-NN output table was processed using the 'directLFQ' package implemented in Python (https://github.com/MannLabs/directlfq) (Ammar et al, 2023) to quantify protein intensities.

### Bioinformatics data analysis

All bioinformatics data analysis was performed in Python (version 3.12.2) within a jupyter notebook (version 7.1.2) environment.

**Calculation of the labeling efficiency:** The MaxQuant output table were filtered for "Reverse", "Only identified by site modification", and "Potential contaminants" before further processing. Labeling efficiency was calculated based on the ratio of summed intensities of labeled precursors relative to all detected precursors.

**Determination of the channel q-value cutoff threshold:** To ensure the false discovery rate (FDR) is maintained below 1%, the distribution of channel $q$ values for each precursor identified in the $\Delta 0$-labeled urine peptides was systematically investigated. This analysis permitted the establishment of a channel q-value cutoff that allows only $\Delta 0$-labeled precursors to be identified, thereby effectively distinguishing between true and false identifications. Subsequently, channel q-value cutoffs ranging from 0.05 to 0.5 at intervals of 0.05 were applied to filter the precursor identifications. The ratio of precursor identifications in the $\Delta 4$ and $\Delta 8$ channels relative to those in the $\Delta 0$ channel was then calculated. This step was crucial for validating the accuracy of the channel q-value cutoff and confirming that the false discovery rate was consistently maintained below 1%.

**Calculation of the coefficients of variation:** For the urine proteomics dataset, analytical CVs were calculated from raw protein intensities in the reference channels across all runs. The median value was reported as the overall analytical coefficient of variation for the entire urine mDIA workflow. For the plasma proteomics dataset, analytical CVs were calculated from raw protein intensities in the pooled QC samples across all plates. The median value was reported as the overall analytical coefficient of variation for the entire plasma workflow. In both cases, biological CVs were calculated from raw protein intensities across participants within each cohort.

**Preprocessing of the urine and plasma proteomics datasets:** Protein intensities were log2-transformed prior to all downstream statistical analyses. For the urine proteomics dataset, stringent filtering was applied by removing all samples that had protein group identifications less than 800. For the plasma proteomics dataset, samples that were outliers based on protein group identifications below the lower threshold (calculated as the 25th percentile minus 1.5 times the interquartile range) were excluded from further analysis. In both cases, batch effects associated with the plate were observed based on clusters in the principal component analysis, which were subsequently corrected using pyCombat.

**Pairwise correlation of urine and plasma proteomics datasets:** Pairwise correlation was performed to assess the correlation between paired urine and plasma protein intensities across the cohort. A Python script based on the open-source packages scipy (version 1.12.0) and statsmodels (version 0.14.0) was developed to determine the Pearson correlation of each commonly identified protein across the cohort between the urine and plasma proteomes, and to control for multiple hypothesis testing. Significance level was controlled at an FDR-adjusted $P$ value by Benjamini–Hochberg correction of <0.01 and an absolute value of Pearson's correlation coefficient $r > 0.4$.

**Principal component analysis of the urine and plasma proteomes:** Principal component analysis (PCA) was employed to reduce the dimensionality of the proteomics data and to investigate the extent to which age contributes to variance in protein expression across the cohort. Missing values in the dataset were imputed using the $k$-nearest neighbors (KNN) method with the default setting of five nearest neighbors, using the sklearn package (version 1.4.2). The imputed dataset was then standardized and scaled prior to PCA execution. The loadings of the first two principal components were analyzed to observe shifting patterns across different ages within the cohort, elucidating age-related differences in protein expression.

**ANOVA and hierarchical clustering analysis of age-related proteins across both sexes:** One-way analysis of variance (ANOVA) was performed using the scipy.stats (version 1.12.0) package to identify proteins with significant expression differences across sex and age groups in plasma and urine samples. Preprocessing included retaining proteins with at least 70% non-missing values. For each protein, groups were formed based on sex and age, and the f_oneway function from scipy.stats was used to compute F-statistics and $P$ values. False discovery rate (FDR) correction using the Benjamini–Hochberg method was applied using the multipletests function from the statsmodels.stats.multitest (version 0.14.0) module to adjust $P$ values for multiple hypothesis testing. Proteins with adjusted $P$ values below the specified threshold ($<0.01$) were considered significant.

The resulting list of significant genes was imported into Perseus (version 2.0.11) for unsupervised hierarchical clustering. Clustering was performed using Euclidean distance as the metric and average linkage. Preprocessing steps included enabling k-means clustering with 300 clusters and a maximum of 10 iterations, with no additional constraints applied.

**ANCOVA of selected diseases in the cohort:** Analysis of covariance (ANCOVA) was conducted using the Python statsmodels package (version 0.14.0) to examine protein expression differences between control and disease cohorts while accounting for age and sex as covariates. ANCOVA models were constructed for each protein with disease status as the categorical predictor and age and sex as covariates using the "ols" function. $P$ values for the pairwise comparison between the mean protein intensity of the control group and the disease group were derived from the model. Mean protein intensities for control and disease groups were calculated, and log2 fold changes (log2FC) were determined. False discovery rate (FDR) correction by Benjamini–Hochberg procedure was applied to adjust $p$ values using the multipletests function from the statsmodels package. Proteins with adjusted $p$ values below the threshold ($<0.05$) were classified as significant.

**SNOMED CT graph and subgraph construction:** To construct the ontology framework, we followed the pipeline illustrated in Fig. EV5A. We began by building a global graph of SNOMED CT concepts using the NetworkX Python package, including all active, non-duplicated terms (439,619 nodes) and their "is a" subtype relationships (1,048,560 edges). Although SNOMED CT is designed as a hierarchical ontology, cycles can still occur due to overlapping classifications or cross-linked concepts. To ensure a clean hierarchical structure suitable for downstream analysis, we implemented a cycle-breaking algorithm using an ensemble of strategies that select edges for removal based on underlying graph hierarchy (Sun et al, 2017). This step removed 1351 cyclic edges, resulting in a directed acyclic graph (DAG) containing 439,619 nodes and 1,047,209 edges.

From this cleaned DAG, we extracted a cohort-specific subgraph to serve as the basis for ontology embedding and disease clustering. To focus our analysis on clinically relevant relationships and reduce computational complexity, we filtered the graph to include only the SNOMED CT terms corresponding to diagnoses present in our pediatric cohort, along with their direct ancestors (up to two levels above) and direct descendants (up to two levels below). This approach was designed to strike a meaningful balance by including enough hierarchical context to avoid overly narrow, disconnected nodes while excluding excessively broad, non-informative parent terms. Using only the diagnosis nodes resulted in a highly fragmented subgraph, which is suboptimal for embedding techniques such as node2vec that rely on local neighborhood structure to learn informative representations. To determine the appropriate inclusion depth, we empirically tested three configurations: one level, two levels, and three levels of ancestry and descent. We evaluated how each affected graph connectivity by computing the maximum depth from each diagnosis node to its most distant root node. Increasing to three levels did not further increase this metric, indicating that additional expansion no longer improved hierarchical depth or node connectivity (Fig. EV5C). Based on this analysis, we selected the two-level window as the optimal tradeoff between interpretability, structural integration, and computational efficiency.

**SNOMED CT vector embedding**: To explore the network's structural properties, we embedded the nodes of the cohort-specific subgraph into a 128-dimensional vector space using the node2vec algorithm implemented in the node2vec Python package (version 0.5.0). The following parameters were used: walk length = 20, number of walks = 20, return parameter $p = 3$, in-out parameter $q = 1.5$, and window size = 10. To ensure reproducibility, we fixed the random seed (seed = 42) and restricted the number of parallel workers to one. These settings generated stable and deterministic embeddings for downstream analysis.

**SNOMED CT clustering**: To cluster the embedded nodes, we tested three clustering methods: $k$-means, agglomerative, and spectral clustering. For each method, we evaluated a range of cluster numbers ($k = 10$–$200$) and computed the Silhouette score using cosine distance. A nonlinear saturation model was then fitted to the Silhouette score curves using a beta-optimized asymptotic function. The optimal number of clusters was defined as the smallest $k$ where the Silhouette score reached 97% of its fitted asymptote. Among the tested methods, $k$-means achieved the best performance with a Silhouette score of 0.31 at $k = 43$, and was selected for downstream analysis (Fig. EV5D). As a result, this clustering strategy substantially mitigated the issue of small patient groups, reducing the number of disease categories with fewer than five patients from 344 to just 8 (Fig. EV5A, right panel).

To assess the biological and clinical coherence of resulting clusters, we used two complementary strategies. First, we performed manual clinical review of each cluster to evaluate the plausibility of disease groupings (Dataset EV4). Second, we calculated pairwise Euclidean distances between embedded diagnosis nodes within each cluster to quantify internal cohesion (Fig. 4A). For visualization, the node2vec embeddings were projected into two dimensions using UMAP (n_neighbors = 30, min_dist = 0.2, random_state = 42) as implemented in the umap-learn package (version 0.5.6). The resulting projection, shown in Fig. 4B, illustrates the spatial organization of disease clusters based on semantic similarity. Cluster annotations were overlaid using median-centered labels for each group.

## Data availability

The mass spectrometry proteomics data have been deposited to the ProteomeXchange Consortium via the PRIDE partner repository with the dataset identifier PXD058960.

The source data of this paper are collected in the following database record: biostudies:S-SCDT-10_1038-S44321-025-00253-z.

## Peer review information

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

## Acknowledgements

We thank all members of the Department of Proteomics and Signal Transduction at the Max Planck Institute of Biochemistry, with special recognition to Patricia Skowronek for assisting with the generation of timsTOF methods and Max Zwiebel for helping establish protocols on the Bravo. We are also grateful to Monika Ludewig and Ute Bossmanns from the OMICs Lab biobank at the Dr. von Hauner Children's Hospital, as well as all clinic staff who contributed to sample and data collection. This project was partially supported by the Max Planck Society for the Advancement of Science, the German Center for Child and Adolescent Health (DZKJ) under the funding registry 01GL2406D, the Eva Mayr-Stihl Stiftung (funding registry 220016), and the Care-for-Rare Foundation (funding registry 190121).

## Author contributions

**Ericka C M Itang**: Conceptualization; Data curation; Software; Formal analysis; Investigation; Visualization; Methodology; Writing—original draft; Writing—review and editing. **Vincent Albrecht**: Formal analysis; Investigation. **Alicia-Sophie Schebesta**: Formal analysis; Investigation. **Marvin Thielert**: Formal analysis; Investigation. **Anna-Lisa Lanz**: Data curation; Project administration. **Katharina Danhauser**: Data curation; Project administration. **Jessica Jin**: Data curation; Project administration. **Tobias Prell**: Data curation; Project administration. **Sophie Strobel**: Data curation; Project administration. **Christoph Klein**: Conceptualization; Resources; Supervision; Funding acquisition; Methodology; Project administration. **Matthias Mann**: Conceptualization; Resources; Supervision; Funding acquisition; Methodology; Project administration; Writing—review and editing. **Susanne Pangratz-Fuehrer**: Data curation; Formal analysis; Supervision; Methodology; Project administration; Writing—review and editing. **Johannes Mueller-Reif**: Conceptualization; Data curation; Formal analysis; Supervision; Validation; Investigation; Visualization; Methodology; Writing—original draft; Project administration; Writing—review and editing.

Source data underlying figure panels in this paper may have individual authorship assigned. Where available, figure panel/source data authorship is listed in the following database record: biostudies:S-SCDT-10_1038-S44321-025-00253-z.

## Funding

## Disclosure and competing interests statement

MM is an indirect shareholder in EvoSep Biosystems. The remaining authors declare no competing interests.

# Expanded View Figures

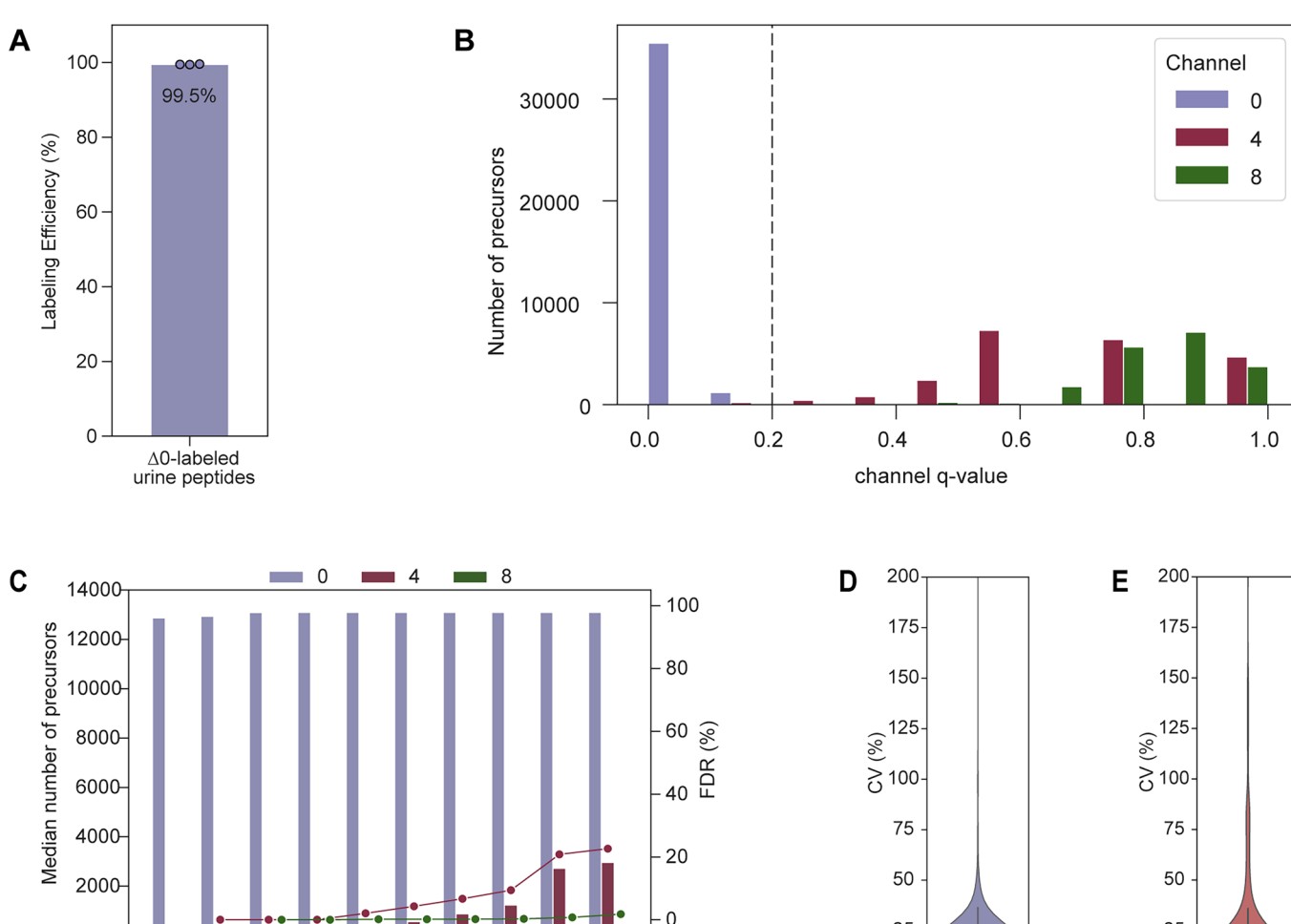

**Figure EV1. Assessment of data quality in the urine and neat plasma proteomics datasets.**

(A) Dimethyl labeling efficiency in the urine dataset, assessed by comparing the intensity ratios of Δ0-labeled peptides to all detected peptides in DDA mode across three technical replicates ($n = 3$). (B) Distribution of precursor identifications for a Δ0-labeled pooled urine sample analyzed through the mDIA workflow across three technical replicates ($n = 3$). (C) Median number of precursor identifications for each of the three dimethyl labeling channels (Δ0/Δ4/Δ8), calculated across three technical replicates for a Δ0-labeled pooled urine sample ($n = 3$). The secondary axis displays the percentage of false discovery rate (FDR), derived from the ratio of false precursor identifications in the Δ4 and Δ8 channels relative to the Δ0 channel. (D) Violin plot of analytical coefficients of variation (%CV) for protein groups identified in the reference channel of the urine proteomics dataset, filtered by a channel q-value < 0.2, across 553 technical replicates ($n = 553$). A horizontal dashed line indicates the median analytical CV of 21%. The internal boxplot shows the interquartile range (IQR), with whiskers extending to the most extreme point within 1.5 times the IQR. Outliers are not shown for clarity. (E) Violin plot of analytical coefficients of variation (%CV) for protein groups identified in the QC samples of the neat plasma proteomics dataset, filtered by a channel q-value < 0.2, across 112 technical replicates ($n = 112$). A horizontal dashed line indicates the median analytical CV of 13%. The internal boxplot shows the interquartile range (IQR), with whiskers extending to the most extreme point within 1.5 times the IQR. Outliers are not shown for clarity.

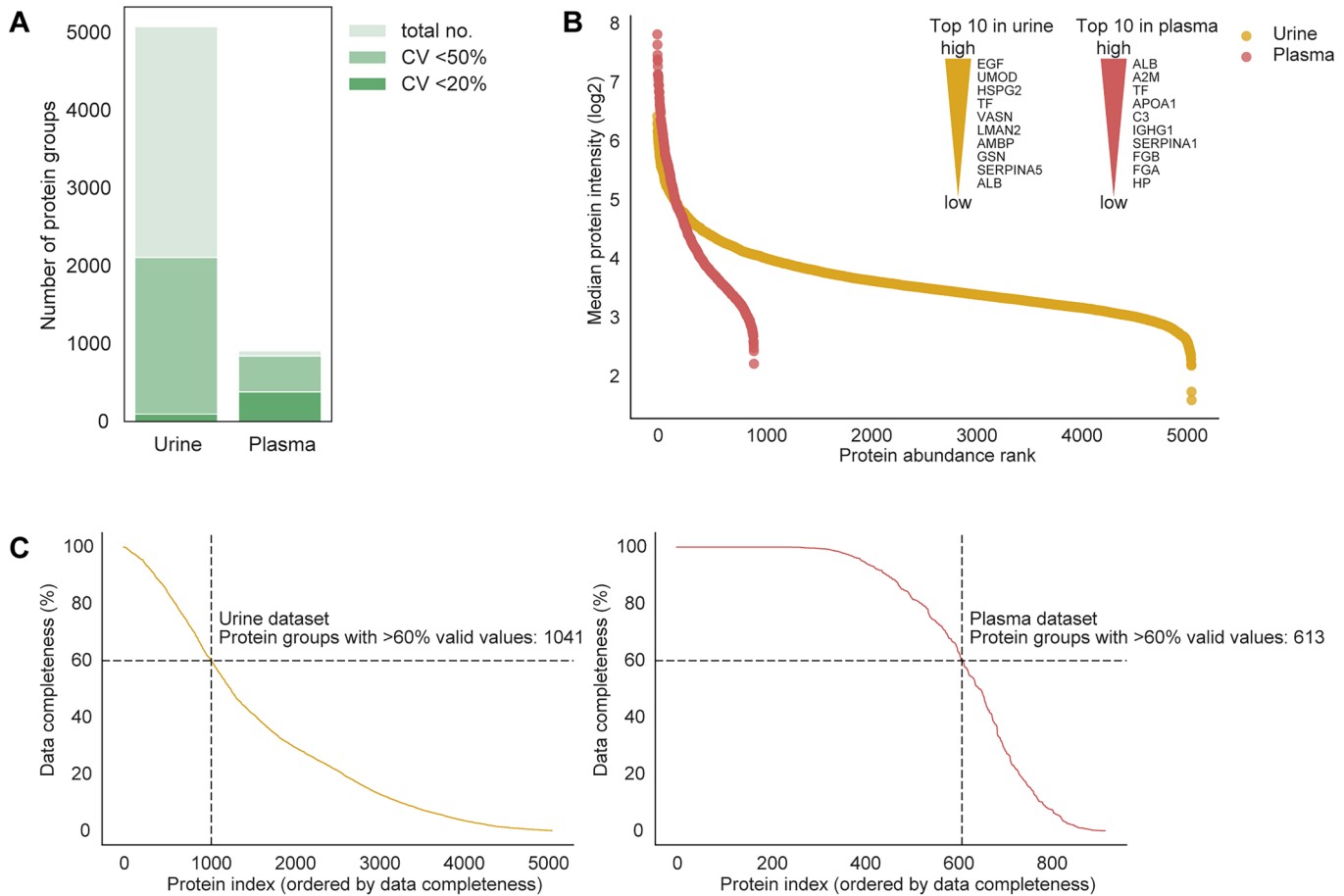

**Figure EV2. Proteomic profiling of urine and neat plasma samples in a pediatric cohort.**

(A) Cumulative protein group identifications in each body fluid. Proportion of protein groups with <50% and <20% biological CV for both is shown. (B) Abundance rank plot of protein groups based on median protein intensities. The top ten most abundant protein groups for each body fluid are listed. (C) Data completeness curve for each body fluid (left: urine; right: neat plasma). Number of protein groups quantified with >60% data completeness is shown.

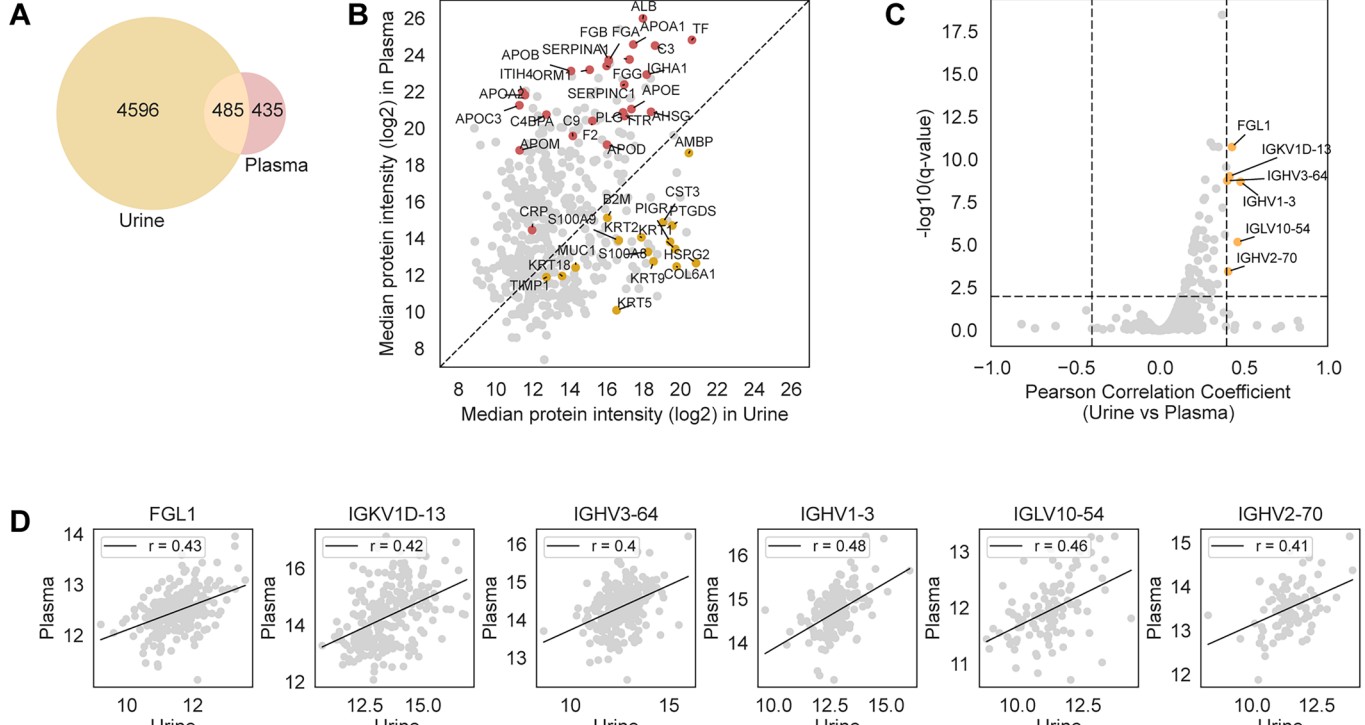

**Figure EV3. Integration of urine and neat plasma proteome data.**

(A) Venn diagram showing distinct protein group identifications in the urine and neat plasma proteomes, or in both. (B) Abundance map of the commonly identified proteins in the urine and neat plasma proteomes, showing the correlation between their median protein intensities in log2 space. Apolipoproteins, complement system proteins, coagulation factors, and other known plasma proteins are highlighted in red. Proteins related to kidney function and filtration, as well as structural and epithelial proteins such as keratins and mucins are highlighted in yellow. A diagonal line representing x = y is shown as a gray, dashed line. (C) Volcano plot displaying Pearson correlation coefficients between protein intensities in the urine and neat plasma proteome across the cohort, along with their associated FDR-adjusted $P$ values (Benjamini–Hochberg correction). Statistically significant, moderately correlating proteins (FDR-corrected $P$ value < 0.01 and Pearson's $r \geq 0.4$) are highlighted. (D) Correlation plots of log2-transformed protein intensities of the urine and neat plasma proteomes for each individual protein highlighted in (C). Regression lines are shown as solid black lines, and the Pearson's $r$ values are displayed.

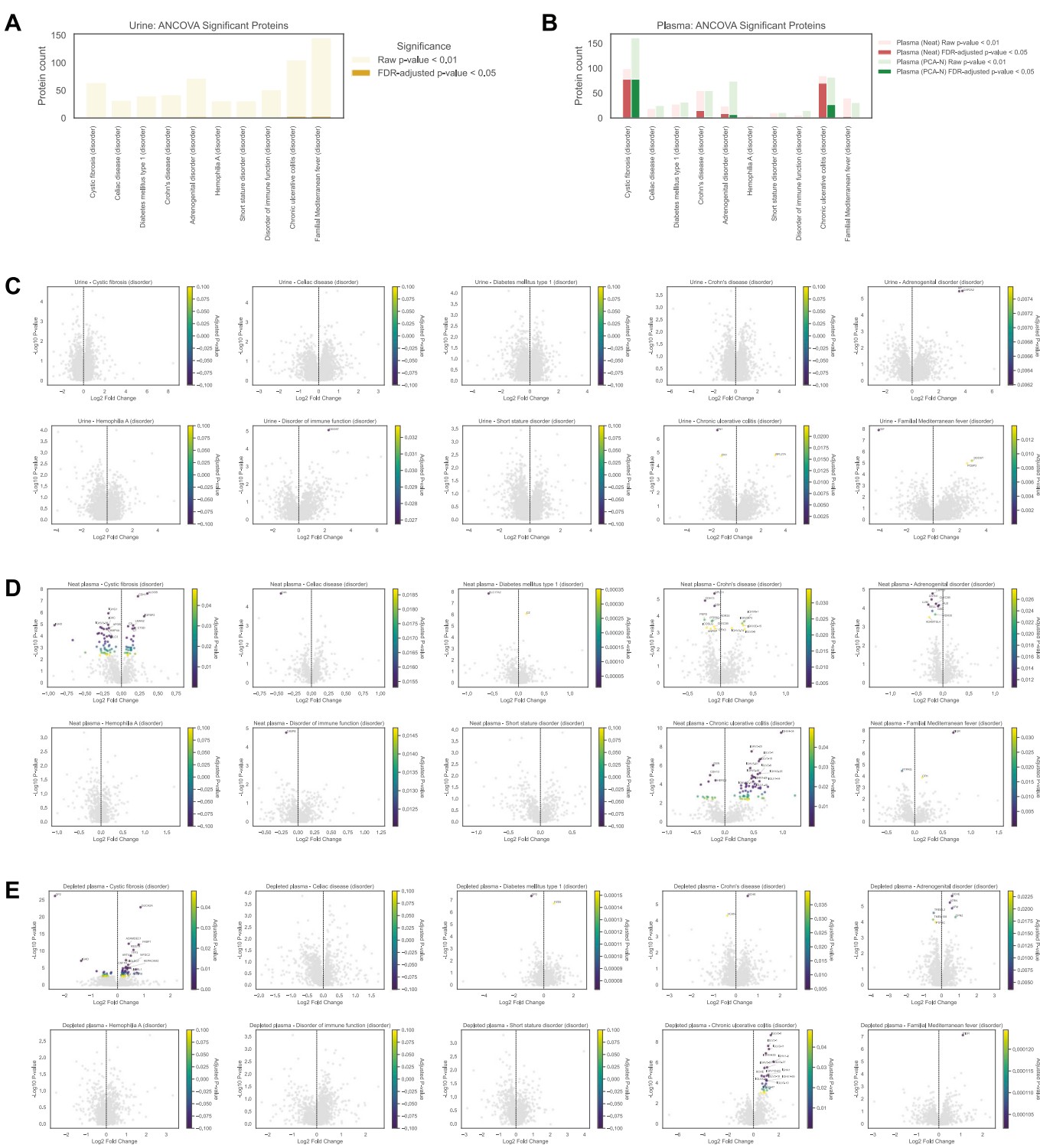

◀ **Figure EV4.   Differential expression analysis of the top ten most prevalent diseases versus healthy controls.**

(**A**) Number of differentially regulated protein groups (raw *P* value < 0.01) in the urine proteome for each disease in the top ten most prevalent diseases, as identified by ANCOVA with age and sex as covariates. Highlighted in darker yellow color is the number of differentially regulated protein groups that have an FDR-adjusted (Benjamini–Hochberg correction) *P* value < 0.05. (**B**) Number of differentially regulated protein groups (*P* value < 0.01) in the plasma proteome for each disease in the top ten most prevalent diseases, as identified by ANCOVA with age and sex as covariates. The neat plasma dataset measured on the Bruker timsTOF HT is shown in red, while the PCA-N plasma dataset measured on the Thermo Orbitrap Astral is shown in green. Highlighted in darker colors are the number of differentially regulated protein groups that have an FDR-adjusted (Benjamini–Hochberg correction) *P* value < 0.05. (**C**) Representative volcano plots illustrating differential protein expression for each disease compared to healthy controls in the urine proteome, analyzed using ANCOVA with sex and age as covariates. Note that the volcano plot for adrenogenital disorder is also shown in Fig. 3E. (**D**) Representative volcano plots illustrating differential protein expression for each disease compared to healthy controls in the neat plasma proteome, analyzed using ANCOVA with sex and age as covariates. Note that the volcano plot for cystic fibrosis is also shown in Fig. 3A. (**E**) Representative volcano plots illustrating differential protein expression for each disease compared to healthy controls in the PCA-N plasma proteome, analyzed using ANCOVA with sex and age as covariates. Note that the volcano plot for cystic fibrosis is also shown in Fig. 3C.

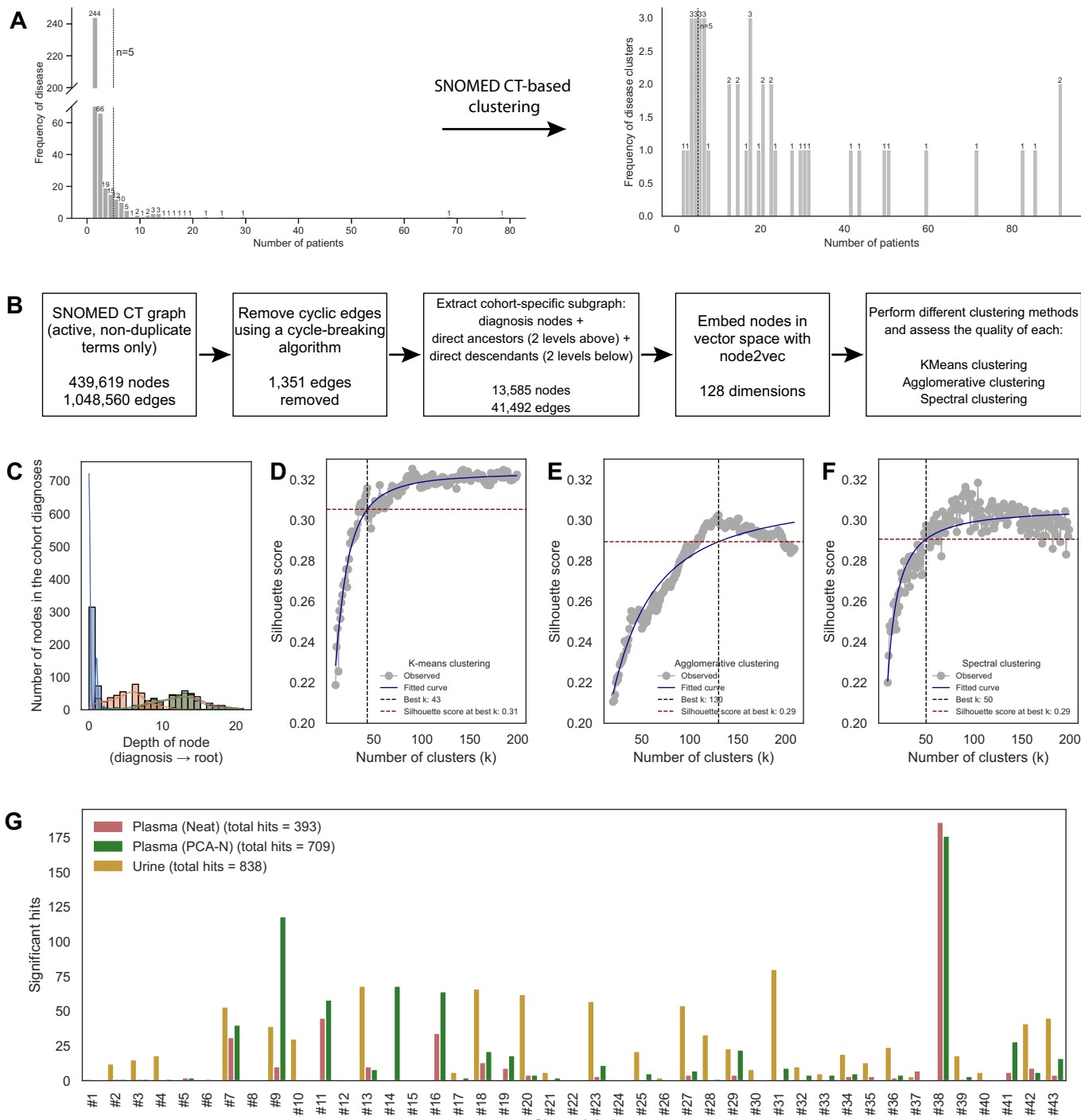

◀  **Figure EV5. Construction and clustering of SNOMED CT ontology network for disease grouping.**

(A) Distribution of patient counts across disease categories before and after SNOMED CT-based clustering. The main histogram shows the number of original disease categories in the cohort (prior to clustering), binned by patient count. The dashed vertical line at $n = 5$ indicates the threshold for identifying the number of disease groups containing less than five patients. The inset displays the corresponding distribution after clustering, demonstrating a marked reduction in the number of small patient groups. (B) Schematic of the SNOMED CT-based clustering pipeline. (C) Depth from diagnosis node to root node across different inclusion windows (1, 2, or 3 levels of ancestors and descendants). For each subgraph, we calculated the maximum upward path length from each cohort diagnosis node to the most distant ancestor. (D) Silhouette score curves across values of k (number of clusters) ranging from 10 to 200 for k-means clustering. The best k was defined as the point where the Silhouette score reached 97% of its maximum. (E) Same as in (D) but for agglomerative clustering. (F) Same as in (D) but for spectral clustering. (G) Total number of significantly altered protein groups identified per body fluid after ontology-guided disease clustering. Bar plots show the number of differentially expressed proteins in each proteomics dataset—urine, neat plasma, and PCA-N plasma—compared to healthy controls. Significance was determined using Welch's $t$ test with Benjamini–Hochberg correction (FDR < 0.05).

