## [Peer Review File · EMBO Molecular Medicine]

Ontology-guided clustering enables proteomic analysis of rare pediatric disorders

Ericka Itang, Vincent Albrecht, Alicia-Sophie Schebesta, Marvin Thielert, Anna-Lisa Lanz, Katharina Danhauser, Jessica Jin, Tobias Prell, Sophie Strobel, Christoph Klein, Matthias Mann, Susanne Pangratz-Führer, and Johannes Müller-Reif

Corresponding author: Johannes Müller-Reif (jomueller@biochem.mpg.de)

Review Timeline:

Submission Date:	18th Dec 24
Editorial Decision:	13th Jan 25
Revision Received:	3rd Apr 25
Editorial Decision:	28th Apr 25
Revision Received:	8th May 25
Accepted:	9th May 25

Editor: Zeljko Durdevic

Transaction Report:

13th Jan 2025

Dear Dr. Müller-Reif,

Thank you for the submission of your manuscript to EMBO Molecular Medicine. We have now received feedback from the three reviewers who agreed to evaluate your manuscript. All three referees recognize interest of the study but also raise important concerns that should be addressed in a major revision. If you would like to discuss further the points raised by the referees, I am available to do so via email or video. Let me know if you are interested in this option.

We would welcome the submission of a revised version within three months for further consideration. Please let us know if you require longer to complete the revision.

I look forward to receiving your revised manuscript.

Yours sincerely,

Zeljko Durdevic

We require:

- 1) A .docx formatted version of the manuscript text (including legends for main figures, EV figures and tables). Please make sure that the changes are highlighted to be clearly visible.
- 2) Individual production quality figure files as .eps, .tif, .jpg (one file per figure). For guidance, download the 'Figure Guide PDF': (<https://www.embopress.org/page/journal/17574684/authorguide#figureformat>).
- 3) A .docx formatted letter INCLUDING the reviewers' reports and your detailed point-by-point responses to their comments. As part of the EMBO Press transparent editorial process, the point-by-point response is part of the Review Process File (RPF), which will be published alongside your paper.
- 4) A complete author checklist, which you can download from our author guidelines (<https://www.embopress.org/page/journal/17574684/authorguide#submissionofrevisions>). Please insert information in the checklist that is also reflected in the manuscript. The completed author checklist will also be part of the RPF.
- 5) Please note that all corresponding authors are required to supply an ORCID ID for their name upon submission of a revised manuscript.
- 6) It is mandatory to include a 'Data Availability' section after the Materials and Methods. Before submitting your revision, primary datasets produced in this study need to be deposited in an appropriate public database, and the accession numbers and

database listed under 'Data Availability'. Please remember to provide a reviewer password if the datasets are not yet public (see <https://www.embopress.org/page/journal/17574684/authorguide#dataavailability>).

12) Author contributions: You will be asked to provide CRediT (Contributor Role Taxonomy) terms in the submission system. These replace a narrative author contribution section in the manuscript.

13) A Conflict of Interest statement should be provided in the main text.

14) Every published paper now includes a 'Synopsis' to further enhance discoverability. Synopses are displayed on the journal webpage and are freely accessible to all readers. They include a short stand first (maximum of 300 characters, including space) as well as 2-5 one-sentences bullet points that summarizes the paper. Please write the bullet points to summarize the key NEW findings. They should be designed to be complementary to the abstract - i.e. not repeat the same text. We encourage inclusion of key acronyms and quantitative information (maximum of 30 words / bullet point). Please use the passive voice. Please attach

these in a separate file or send them by email, we will incorporate them accordingly.

15) Include a Reagents and Tools Table as part of the Methods section, which can be downloaded from our author guidelines (<https://www.embopress.org/page/journal/17574684/authorguide#structuredmethods>)

***** Reviewer's comments *****

Referee #1 (Comments on Novelty/Model System for Author):

This is a beautiful piece of work which solved or at least most closely solved the rare diseases. The identification rate of proteome is high. But authors seem to contribute the success of the whole strategy mainly to the analyzing part. Authors should realize that the analyzing can only build on the comprehensive identification of proteome data that truly sensitively reflecting the pathophysiology of the disease. In this case urine contributed 5000 pieces of information compare to 900 from plasma. Authors also didn't know there was a theory that urine reflects early and sensitive changes of the disease than blood, or forgot to address it. Overall, this work is changing the way we study rare diseases and is generalizable to common diseases too.

Referee #1 (Remarks for Author):

This is a beautiful piece of work which solved or at least most closely solved the rare diseases. The identification rate of proteome is high. But authors seem to contribute the success of the whole strategy mainly to the analyzing part. Authors should realize that the analyzing can only build on the comprehensive identification of proteome data that truly sensitively reflecting the pathophysiology of the disease. In this case urine contributed 5000 pieces of information compare to 900 from plasma. Authors also didn't know there was a theory that urine reflects early and sensitive changes of the disease than blood, or forgot to address it. Overall, this work is changing the way we study rare diseases and is generalizable to common diseases too.

Referee #2 (Remarks for Author):

This review understands the importance of identifying molecular targets in rare pediatric diseases.

The authors seemed to have achieved relatively good reproducibility for MS according to EV1.

Unfortunately, the major proteins identified are relatively abundant molecules, such as immunoglobulins, as expected. As shown by the authors, the urine samples seemed to have more information.

The authors claimed the introduction of GO system for the analysis. The method may be a novel one, however, the results discussed in EV3 sound indifferent from the conventional method.

In EV4, the authors focused on cystic fibrosis and IBD. The reviewer understand that the authors focused on these diseases because of the relative abundance in numbers. For the same reasons, the information for these diseases is abundant. The authors seem to find the difficulty to identify novel results. From the title, the reviewer expected a development of study method for more rare diseases, which seemed not to be achieved in this study.

The most critical point is lack of internal standard for urine samples. When assessing urine samples, intrinsic control is important, since urine sample is subject to hydration status. The analysis of urine data is difficult to implement.

Referee #3 (Comments on Novelty/Model System for Author):

Itang et al carried out extensive proteomics analysis on samples derived from a dataset of pediatric rare disease patients and controls. In addition to the traditional analysis, they set out to use the available medical disease ontology to "collapse" individual rare disease diagnosis to disease groups, thereby allowing this data to be analysed using more extensive statistical methods.

This work is a much needed step towards narrowing the data and research gap between rare and more common complex diseases, as well as using tools of bioinformatics in the clinical setting. The publication would benefit from some additional quantitative analysis to illustrate the novelty of their approach, the gain in understanding of these rare conditions as well as placing their work into the broader clinical and therapeutic context.

In particular, general quantitative measurements on how their approaches improved the status quo - such as how many rare disease patients can now be analysed with standard proteomics analysis vs before. The authors should also quantify the within-group similarity of patient groups that they get, as well as give a quantitative measure on how well their chosen ontology covered their patient cohort.

They should also expand their methods to include more details on the extraction of patient data as well as the ontology-based methods.

Referee #3 (Remarks for Author):

Itang et al carried out extensive proteomics analysis on samples derived from a dataset of pediatric rare disease patients and controls. In addition to the traditional analysis, they set out to use the available medical disease ontology to "collapse" individual rare disease diagnosis to disease groups, thereby allowing this data to be analysed using more extensive statistical methods. This work is a much needed step towards narrowing the data and research gap between rare and more common complex diseases, as well as using tools of bioinformatics in the clinical setting.

The publication would benefit from some additional analysis to illustrate the novelty of their approach, the gain in understanding of these rare conditions as well as placing their work into the broader clinical and therapeutic context.

Major comments:

Ma1: Although the title of the paper reads "Ontology-guided clustering enables proteomic analysis of rare pediatric disorders", we are introduced to this step in the analysis in the last part of the paper, without any substantial analysis on what effect this new way of grouping patients has on subsequent analysis. The authors should quantify the improvement gained by clustering. How many patients could be analysed with this approach vs not using SNOMED ct-based clustering?

Ma2: How will this approach impact future clinical care or research? What additional analysis techniques are now feasible? Can we now better diagnose these patients, do we know more about their prognosis? The authors should illustrate this with an example as well as discuss the possibilities in the discussion.

Ma3: Methods section is lacking regarding the extraction of patient data, and the ontology/network based methods to extract patient clusters.

Minor comments:

Mi1: The authors used both urine and plasma proteome to extract clusters of signatures. How does this classification reflect on each patient? Do urine and plasma clusters co-occur? Meaning, are there patients/patient groups who based on one type of sample more similar, vs more dissimilar based on the other type of proteome?

Mi2: The authors used to obtain the ontology-based disease clusters. Have they compared other types of clustering? Louvain? Hierarchical classification based on the feature matrix, other types of community detection ect...

Mi3: " These clusters likely represent biologically meaningful groupings of related SNOMED CT terms, reflecting shared clinical or functional characteristics." It would be crucial to give some quantitative measure to assess clinical similarity within clusters.

Mi4: The authors highlight possible gaps in SNOMED ct_ ", reflecting a single patient and a single SNOMED CT term, likely corresponding to an extremely rare disease or a limitation in the ontology's annotation. " How many of the clusters are single disease -disease annotation clusters? How many patients could not have been mapped to SNOMED IDs at all? How does SNOMED CT compare to other disease nomenclatures and classifications?

Point-by-point response to reviewer comments for “Ontology-guided clustering enables proteomic analysis of rare pediatric disorders”

We thank the reviewers for their detailed and constructive feedback, which has greatly helped us refine and strengthen our manuscript. We are pleased that the reviewers recognized the significance of our study and provided insightful comments to improve its clarity, depth, and rigor.

Reviewer 1 was overall very positive about our work. Their comments focused on aspects that were previously underemphasized in the manuscript, such as the importance of proteomic depth in enabling our analytical framework, as well as points that were previously overlooked, such as the theory that urine may serve as a sensitive indicator of early disease changes compared to blood.

Reviewer 2 acknowledged the relevance of identifying molecular targets in rare pediatric diseases and appreciated the reproducibility of our MS workflows. However, they raised concerns regarding our limited identification of novel results for very rare diseases, asked for the use of internal standards for urine proteomics.

Reviewer 3 was also generally positive and recognized our ontology-guided clustering approach as an important step toward bridging the gap between rare and common diseases. Their primary concern was the need for more quantitative evaluation of our method's impact, specifically on how many rare disease patients became analyzable due to clustering, compared to traditional approaches. Additional suggestions included quantifying within-cluster similarity, evaluating ontology coverage, expanding methodological details, and discussing broader clinical implications.

In response to these comments, we have significantly developed and improved the manuscript during the three-month revision period. Notably, we implemented three major updates:

1. **Expanded plasma proteome coverage** by complementing our neat plasma dataset with data acquired using a perchloric acid-based depletion strategy (PCA-N) (Albrecht et al., 2025), adapted from the workflow developed by the Hanno Steen group (Viode et al., 2023). This strategy more than doubled the total number of protein group identifications – from approximately 900 in neat plasma to over 1,900 in PCA-N plasma – and enriched for low-abundance proteins. As illustrated in the abundance rank plot below (**Response Figure 1**), the PCA-N plasma dataset achieved significantly greater proteomic depth than neat plasma.

Response Figure 1. Abundance rank plot of protein groups based on median protein intensities for the three different body fluids (urine, neat plasma, and PCA-N plasma). Selected low-abundance cytokine-related proteins, as previously characterized by Anderson & Anderson (2002), are annotated. Duplicate annotations reflect instances where a single protein is represented in multiple protein groups.

With the addition of this complementary plasma data, we have enabled the identification of more protein groups that are significantly altered in the disease cluster when compared with the healthy controls (**Response Figure 2**). This has been supplemented into the main manuscript (clean version) as Fig. EV5G and its corresponding text in p. 17, lines 16-22.

“Using this clustering framework, we identified a total of 838 significantly altered protein groups in urine, 393 in neat plasma, and 709 in PCA-N plasma (Fig. EV5G). Clusters with no significant hits in a given body fluid may reflect either the absence of patient samples for that fluid or the lack of statistically significant differences compared to healthy controls. Notably, the number of significant proteins detected in PCA-N plasma was nearly double that of the neat plasma dataset, further underscoring the value of the perchloric acid-based depletion strategy in enhancing proteomic depth and sensitivity.”

Response Figure 2. Total number of significantly altered protein groups identified per body fluid after ontology-guided disease clustering. Bar plots show the number of differentially expressed proteins in each proteomics dataset—urine, neat plasma, and PCA-N plasma—compared to healthy controls. Significance was determined using Welch’s *t*-test with Benjamini-Hochberg correction ($FDR < 0.05$).

2. **Refined the SNOMED CT-based clustering approach** by expanding the cohort-specific subgraph to include two levels of direct ancestors and descendants for each diagnosis node. This adjustment improved the performance of the vector embedding process by increasing graph connectivity and enabling deeper contextual representation. We also tested different clustering approaches (k-means clustering, agglomerative clustering, and spectral clustering) to ensure that we obtain the best clustering quality.
3. **Added multiple qualitative and quantitative metrics** to assess the effectiveness of SNOMED CT-based clustering. These include the Silhouette score to evaluate cluster separation, a manual domain-expert review of biological and clinical coherence within each cluster, and calculation of pairwise Euclidean distances between nodes within each cluster.

In the following point-by-point response, we address each of the reviewers’ concerns in detail, outlining the changes made to the manuscript and the rationale behind them. To clearly indicate these revisions, we have used the following color code:

Black: Reviewer comments
 Blue: Our explanation and response
 Green: Text added or modified in the revised manuscript

We believe these revisions have significantly enhanced the quality of our work and hope that the manuscript now meets the standards for publication in *EMBO Molecular Medicine*. Thank you for your consideration.

Reviewer #1

Comment:

This is a beautiful piece of work which solved or at least most closely solved the rare diseases. The identification rate of proteome is high. But authors seem to contribute the success of the whole strategy mainly to the analyzing part. Authors should realize that the analyzing can only build on the comprehensive identification of proteome data that truly sensitively reflecting the pathophysiology of the disease. In this case urine contributed 5000 pieces of information compare to 900 from plasma. Authors also didn't know there was a theory that urine reflects early and sensitive changes of the disease than blood, or forgot to address it. Overall, this work is changing the way we study rare diseases and is generalizable to common diseases too.

Response:

We thank the reviewer for their positive assessment of our work. Indeed, the identification rate of the proteome is high, particularly in urine that provided 5,000 quantified proteins. We fully acknowledge that the success of our approach is built upon this high identification rate, and this further underscores the efficiency of our streamlined proteomics workflows. To add to this, we also incorporated newly measured plasma proteomics data using the PCA-N strategy, which more than doubled the proteomic depth of our neat plasma dataset and enriched for low-abundance proteins.

These points are now explicitly stated in the revised Discussion of the clean manuscript (p. 21, lines 3–10):

“A key factor contributing to the success of this ontology-based clustering approach lies in the depth of our proteomic data. Through the implementation of a perchloric acid-based depletion strategy, we more than doubled the number of protein group identifications in plasma and enriched for low-abundance proteins, as expected (Albrecht et al, 2025). This expanded proteomic coverage provided the resolution necessary to detect meaningful molecular signatures within the aggregated clusters, allowing us to draw clinically relevant conclusions from grouped rare diseases. These findings highlight the synergistic potential of combining deep molecular data from proteomics with clinically informed ontological structures.”

Additionally, we have incorporated a discussion of the theory that urine may reflect early and sensitive changes in disease pathophysiology compared to blood plasma. This perspective, supported by existing literature (Zhang *et al*, 2022), is now included in the Discussion section of the clean manuscript (p. 20, lines 29-31):

“Additionally, urine is uniquely sensitive to changes in the early stages of disease due to the lack of homeostatic mechanisms that buffer subtle proteomic shifts, unlike plasma, which is tightly regulated (Zhang et al, 2022).”

Reviewer #2:

Comment:

This review understands the importance of identifying molecular targets in rare pediatric diseases.

Response:

We thank the reviewer for their thoughtful feedback and appreciate their recognition of the importance of identifying molecular targets in rare pediatric diseases. Below, we address each of the points raised.

Major comments:

1. The authors seemed to have achieved relatively good reproducibility for MS according to EV1.

We thank the reviewer for recognizing the high precision of our proteomics data. Indeed, we achieved good reproducibility for both the multiplexed-DIA (mDIA) workflow used for the urine dataset and the label-free DIA workflow used for the undepleted (neat) plasma dataset. This is reflected in the minimal analytical coefficients of variation (21% for the urine dataset and 13% for the neat plasma dataset) observed across the entire measurement period, as highlighted in Figure EV1.

In addition, we complemented our neat plasma dataset with measurements from a perchloric acid-based depletion strategy (PCA-N) to enrich for low-abundance proteins. While this dataset exhibited a higher median analytical CV of 29%, this is expected due to the nature of depletion strategies, which disproportionately affect the measurement of high-abundance proteins. Importantly, the enriched low-abundance proteins remain consistent and reproducible (Albrecht *et al*, 2025).

2. Unfortunately, the major proteins identified are relatively abundant molecules, such as immunoglobulins, as expected. As shown by the authors, the urine samples seemed to have more information.

We acknowledge that we identified the relatively abundant molecules, particularly immunoglobulins, when analyzing the moderately correlating proteins between urine and plasma. This is expected given their natural abundance in plasma and their passage through glomerular filtration into urine.

With the addition of our PCA-N plasma dataset, we have now doubled our proteomics depth and enriched for low-abundance proteins. With this complementary dataset, we were able to identify several biologically relevant low-

abundance proteins, including cytokine-related proteins such as CD14, CD44, and CSF1R, as shown in Response Figure 1.

Notably, these low-abundance proteins were already detectable in the neat plasma proteome. Moreover, they contributed meaningfully to biological interpretation, as discussed in Module 3 of the hierarchical clustering analysis of age- and sex-dependent neat plasma proteome. This module exhibited a downward trend in protein intensities across age for both males and females, potentially reflecting the transition from innate immunity to adaptive immunity during childhood development. The presence of cytokine-related proteins such as CD14, CD44, and CSF1R within this module supports this interpretation, given their known roles in innate immune responses. To incorporate these points into the manuscript, we have included a discussion of these examples in the clean version of the revised text (p. 9, lines 4-8):

“Notably, low-abundance plasma proteins involved in cytokine activity, such as the monocyte differentiation antigen (CD14), the cell-surface receptor (CD44), and the macrophage colony-stimulating factor 1 receptor (CSF1R), are included in this module. These proteins play key roles in innate immune responses and demonstrate the capability of our workflow to detect biologically meaningful low-abundance proteins.”

3. The authors claimed the introduction of GO system for the analysis. The method may be a novel one, however, the results discussed in EV3 sound indifferent from the conventional method.

We thank the reviewer for appreciating the novelty of our method. We understand that some confusion may have arisen regarding the role of the Gene Ontology (GO) system in our study, and we would like to clarify this in detail.

The GO system was used solely to identify over-represented biological processes within the modules (or clusters) derived from unsupervised hierarchical clustering of significantly different proteins (FDR-adjusted ANOVA p-value < 0.01) across age and sex (Figure 3). This analysis aimed to explore how the proteome varies with developmental stage and biological sex. As correctly noted by the reviewer, this constitutes a conventional analysis and is not the novel component of our study.

The novel aspect of our work lies in our ontology-guided clustering approach, which uses the Systematized Nomenclature of Medicine Clinical Terms (SNOMED CT) to systematically group rare disease diagnoses based on their semantic and hierarchical relationships. By embedding the SNOMED CT disease terms into a vector space and applying clustering algorithms, we were able to define clinically meaningful “disease clusters” that integrate seamlessly with our proteomics data for differential expression analysis.

We recognize that the use of the term “ontology” may have contributed to the confusion, as both GO and SNOMED CT are ontological frameworks, although with different scopes and applications. In this manuscript, our reference to “ontology-based clustering” specifically pertains to SNOMED CT and its utility in grouping clinically related rare diseases.

The key innovation of this approach is its ability to aggregate clinically related rare pediatric diseases into statistically analyzable clusters, thereby overcoming the inherent challenges of small sample sizes for individual rare conditions. This clustering strategy allows us to derive biological insights that would otherwise be unattainable for individual rare diseases. We hope this clarification highlights the distinction between our conventional use of GO analysis and the novel use of SNOMED CT-based clustering.

4. In EV4, the authors focused on cystic fibrosis and IBD. The reviewer understand that the authors focused on these diseases because of the relative abundance in numbers. For the same reasons, the information for these diseases is abundant. The authors seem to find the difficulty to identify novel results. From the title, the reviewer expected a development of study method for more rare diseases, which seemed not to be achieved in this study.

We thank the reviewer for this important observation. Figure EV4 (and Figure 3) indeed highlights cystic fibrosis and other diseases among the ten most prevalent conditions in our cohort. Our intention in presenting these examples was to demonstrate the “classical” approach to proteomics analysis, where sufficiently large sample sizes allow the use of conventional statistical methods. These served as familiar benchmarks for evaluating proteomic variability and disease-specific signatures in well-powered groups.

However, the main innovation of our study is introduced in the latter part of the manuscript, where we shift our focus to address the more challenging analysis of rare pediatric diseases. In such cases, small patient numbers severely limit the use of traditional statistical tools. To overcome this, we developed an ontology-guided clustering approach using the SNOMED CT network.

As detailed in the manuscript, this method leverages the “is a” subtype relationships within SNOMED CT to embed disease terms into a vector space, followed by k-means clustering to form clinically meaningful disease groups. The number of clusters ($k = 43$) was selected based on maximizing the Silhouette score, ensuring optimal separation and cohesion of disease groupings.

To illustrate the utility of this approach, we focused on diseases in our cohort represented by fewer than five patients. As shown in Figure EV5A, 344 diseases fall into this category, including 244 with only a single patient. In such cases, statistical analysis is infeasible due to a lack of within-group variance. However, by

clustering these diseases with other clinically related terms, we were able to increase the effective sample size and enable meaningful comparisons with healthy controls. This is reflected in the inset of Figure EV5A, where the number of underpowered disease categories dropped from 344 to just 8 after clustering. This has been explained in detail in p. 15, lines 4-14 of the clean revised manuscript:

“To explore the network's structural properties, we embedded the nodes of the cohort-specific subgraph into a 128-dimensional vector space using the node2vec algorithm. We then applied three clustering approaches to group the node embeddings: k-means clustering, agglomerative clustering, and spectral clustering. For each method, we systematically evaluated a range of cluster numbers ($k = 10$ to 200) and used the Silhouette score to assess clustering quality (Fig. EV5D-F). The optimal number of clusters was defined as the point at which the fitted Silhouette score curve reached 97% of its maximum value. Among the methods tested, k-means clustering achieved the highest performance, achieving a Silhouette score of 0.31 at $k = 43$ (Fig. EV5D). As a result, this clustering strategy substantially mitigated the issue of small patient groups, reducing the number of disease categories with fewer than five patients from 344 to just 8 (Fig. EV5A, right panel).”

Most importantly, this approach allowed us to uncover biologically relevant proteomic signatures that would have otherwise been inaccessible. For example, in Cluster 18 of the urine proteome, we grouped four diseases, each represented by only one patient in our cohort: Congestive cardiomyopathy (disorder), Hypoplastic left heart syndrome (disorder), Long QT syndrome (disorder), and Transplanted heart present (disorder). Without clustering, differential expression analysis would not have been possible. However, through our framework, we identified 66 statistically significant protein differences in this group compared to healthy controls (Fig. 4E). This demonstrates the core strength of our method: enabling the proteomics analysis of rare diseases that would be individually underpowered for statistical analysis. All of these points have been discussed in detail in the last section of the Results in the clean manuscript (p. 16, lines 23-36):

“Building on these observations, we next tested whether these low-CV clusters exhibited distinct proteomic signatures compared to healthy controls. To address unequal variance and unbalanced sample sizes between the groups, we applied Welch's t-test for differential expression analysis. In Cluster 18 of the urine proteome dataset, this approach identified 66 protein groups with significantly altered expression relative to controls (Fig. 4E), using an FDR-adjusted p-value threshold (Benjamini-Hochberg correction) of < 0.05 . Among the upregulated proteins, ceruloplasmin (CP), an independent predictor of long-term all-cause mortality in patients with heart failure (Hammadah et al, 2014), and thioredoxin (TXN), a redox regulator previously linked to chronic heart failure (Jekell et al, 2004; Sánchez-Villamil et al, 2016), are particularly

notable. Conversely, protein kinase C delta (PRKCD) was significantly downregulated. Given PRKCD's role in cardiomyocyte apoptosis and oxidative stress signaling and its association with adverse cardiac remodeling (Miao et al, 2022), its downregulation in this cluster is unexpected. It may reflect distinct regulatory dynamics in pediatric structural heart disease, warranting further investigation.”

5. The most critical point is lack of internal standard for urine samples. When assessing urine samples, intrinsic control is important, since urine sample is subject to hydration status. The analysis of urine data is difficult to implement.

We fully agree with the reviewer that the analysis of urine is inherently variable because its composition is influenced by various physiological and external factors such as hydration status. To address this, standardization strategies like creatinine normalization are commonly used in routine clinical tests to correct for urine concentration variability.

This study employed a discovery-based untargeted proteomics approach, and we normalized our data based on the total protein content of each urine sample. Specifically, we standardized the peptide input by ensuring that 100 ng of peptides was injected per sample. This approach reduces variability due to protein concentration differences and ensures comparability across samples prior to mass spectrometry. This aligns with established practices in proteomics, including those described in a previous study from our lab (Winter et al., 2021).

In conclusion the here applied protein input-normalized approach fundamentally reduces the need for internal standards. The intrinsic problem that urine is a less-well controlled body fluid as for examples plasma (where total protein levels are quite constant) is addressed by two steps in the workflow: (i) input normalization, meaning the loading of equal amounts of peptides into the mass spectrometer and (ii) the application of label free protein quantification upon normalization to total peptide quantity within each LCMS run for each quantified precursor. This ensures that the intensity value we compare in our data for each protein of each sample is a representation of the protein quantity per total protein amount.

Reviewer #3:

Itang et al carried out extensive proteomics analysis on samples derived from a dataset of pediatric rare disease patients and controls. In addition to the traditional analysis, they set out to use the available medical disease ontology to "collapse" individual rare disease diagnosis to disease groups, thereby allowing this data to be analysed using more extensive statistical methods. This work is a much-needed step towards narrowing the data and research gap between rare and more common complex diseases, as well as using tools of bioinformatics in the clinical setting. The publication would benefit from some additional quantitative analysis to illustrate the novelty of their approach, the gain in understanding of these rare conditions as well as placing their work into the broader clinical and therapeutic context.

In particular, general quantitative measurements on how their approaches improved the status quo - such as how many rare disease patients can now be analysed with standard proteomics analysis vs before. The authors should also quantify the within-group similarity of patient groups that they get, as well as give a quantitative measure on how well their chosen ontology covered their patient cohort.

They should also expand their methods to include more details on the extraction of patient data as well as the ontology-based methods.

We thank the reviewer for their thoughtful and encouraging feedback, and we are pleased that they recognized the significance and translational potential of our ontology-guided framework for analyzing rare pediatric diseases. We agree that additional quantitative measures and clearer exposition of our methodology help to illustrate the novelty and impact of our approach. In response to these suggestions, we have made several key improvements to the revised manuscript, including the expansion of our plasma proteomics dataset to include both undepleted, neat plasma and perchloric acid-depleted plasma, refining the SNOMED CT-based clustering approach, and adding multiple qualitative and quantitative metrics. We describe more of these in detail in the responses below.

Major comments:

1. Although the title of the paper reads "Ontology-guided clustering enables proteomic analysis of rare pediatric disorders", we are introduced to this step in the analysis in the last part of the paper, without any substantial analysis on what effect this new way of grouping patients has on subsequent analysis. The authors should quantify the improvement gained by clustering. How many patients could be analysed with this approach vs not using SNOMED ct-based clustering?

We appreciate the reviewer's suggestion to more clearly quantify the benefits of clustering patients based on SNOMED CT. In the revised manuscript, we now explicitly report that 344 disease categories in our cohort were represented by fewer than five

patients, with 244 of these represented by a single patient (Fig. EV5A). By applying SNOMED CT-based clustering, we reduced the number of underpowered disease groups to just 8 (Fig. EV5A, inset), thereby enabling statistical comparisons that would not have been feasible without clustering. We emphasize this in both the Results (p. 15, lines 12–14) and the Discussion (p. 20, lines 35-38) as a key demonstration of how our approach enhances analytical power.

p.15 lines 12-14:

“As a result, this clustering strategy substantially mitigated the issue of small patient groups, reducing the number of disease categories with fewer than five patients from 344 to just 8 (Fig. EV5A, right panel).”

p. 20, lines 35-38:

“The power of our integrated approach is particularly evident in how it handles rare conditions. By using network-based clustering of SNOMED CT terms and concepts, we could analyze conditions with only a single patient by connecting them to biologically related disorders, thereby gaining statistical power while maintaining clinical relevance.”

2. How will this approach impact future clinical care or research? What additional analysis techniques are now feasible? Can we now better diagnose these patients, do we know more about their prognosis? The authors should illustrate this with an example as well as discuss the possibilities in the discussion.

We thank the reviewer for raising this important question regarding the broader implications of our approach. Our ontology-guided clustering framework represents a methodological advancement that can facilitate the study of rare diseases in a scalable and clinically interpretable manner. By clustering SNOMED CT terms together based on their semantic relationships, we are able to group together clinically similar but individually underpowered conditions. This makes it feasible to apply statistical methods that would otherwise be inaccessible for single-patient diseases.

As an illustrative example, Cluster 18 of the urine proteome grouped four structural heart diseases (Congestive cardiomyopathy, Hypoplastic left heart syndrome, Long QT syndrome, and Transplanted heart present), each represented by a single patient in our cohort. On their own, these conditions would not allow any statistical testing. However, after clustering, we were able to identify 66 significantly altered protein groups compared to healthy controls (Fig. 4E), including proteins with known relevance to cardiac pathology such as ceruloplasmin (CP) and thioredoxin (TXN). These findings may inform biomarker discovery efforts and contribute to a better molecular understanding of pediatric cardiac conditions.

Beyond this, our clustering framework allows us to generate new hypotheses about shared (or divergent) biological pathways across clinically related but traditionally

separate conditions. For example, one of the clusters that we presented in our manuscript is Cluster 12, which consists of SNOMED CT terms involving autoimmune and allergic disorders. An analysis of this cluster can prompt exploration of common and overlapping immunological mechanisms.

Finally, identifying convergent molecular patterns within disease clusters may support drug repurposing strategies by highlighting shared therapeutic targets across related conditions.

3. Methods section is lacking regarding the extraction of patient data, and the ontology/network-based methods to extract patient clusters.

We thank the reviewer for this important information. We have now expanded the Methods section to detail our patient data extraction process, including what data were collected, how they were obtained and verified, and how they were stored and standardized. Specifically, we have added the following information:

- **Patient data collected:** We extracted comprehensive clinical and demographic data for each patient, including age, sex, height, weight, vital signs, imaging results, and all clinical diagnoses. We also documented each patient's relevant clinical features and symptoms in detail to inform downstream phenotypic analysis.
- **Data sources and collection methods:** All patient-related data and diagnoses were collected from the hospital's electronic health record system (KAS) and imported into the CentraXX software platform, which served as our centralized data repository. From CentraXX, all relevant clinical data were extracted for the study and manually curated by our research team to ensure accuracy and completeness. Supplementary clinical details were also obtained via structured questionnaires (LimeSurvey).
- **Diagnosis verification:** All extracted diagnoses were cross-checked against the original medical records and then confirmed by the patients' treating pediatric specialists. This verification step was done to ensure that every recorded diagnosis was accurate and up-to-date, providing a high-quality clinical dataset for analysis.
- **Data standardization:** To maintain consistency across the cohort, we standardized all clinical information using established medical ontologies and coding systems. Diagnoses were encoded using SNOMED CT and ICD-10 terminology, with Orpha numbers applied for rare diseases. Medication data were recorded and categorized according to the Anatomical Therapeutic Chemical (ATC) classification. In addition, we formally annotated each patient's phenotypic abnormalities using Human Phenotype Ontology (HPO) terms, allowing for a structured and precise representation of clinical features.

- **Pseudonymization and data storage:** All patient-related data were pseudonymized prior to analysis. Each participant was assigned a unique study identifier, and any directly identifying information was removed from the research dataset. The pseudonymized clinical data are stored securely on the hospital's servers, accessible only to authorized study personnel. This process complies with institutional data protection protocols and ethical guidelines, ensuring patient confidentiality and data security.

These are all added into p. 24, lines 20-34 of the Method section:

Clinical data extraction and management

Clinical and demographic data—including age, sex, height, weight, vital signs, imaging results, and diagnoses—were collected for each participant from the hospital's electronic health record system (KAS) and transferred into the CentraXX software platform, which served as the central repository for clinical study data. All relevant patient data were extracted from CentraXX and manually curated by trained researchers to ensure completeness and accuracy. Diagnoses were reviewed and verified by the treating pediatric specialists and encoded using standardized medical terminologies (ICD-10 and SNOMED CT), with Orpha codes applied for rare conditions. Medications were recorded according to the Anatomical Therapeutic Chemical (ATC) classification. To enable structured representation of clinical features, phenotypic abnormalities were annotated using the Human Phenotype Ontology (HPO). Additional clinical information not captured in the EHR was collected via structured electronic questionnaires (LimeSurvey). All clinical data were pseudonymized and stored securely at the hospital, accessible only to authorized study personnel in compliance with data protection regulations.

We also expanded our Methods section to include a more comprehensive description of the ontology-based clustering approach (p. 32-33):

SNOMED CT graph and subgraph construction

To construct the SNOMED CT-based disease network, we parsed the official SNOMED CT release files and built a global graph using the NetworkX Python package (version 3.3). The graph included all active, non-duplicated clinical concepts (439,619 nodes) and their "is a" subtype relationships (1,048,560 directed edges). To ensure a clean hierarchical structure, we applied a cycle-breaking algorithm that uses an ensemble of strategies to identify and remove 1,351 cyclic edges (Sun et al, 2017), yielding a directed acyclic graph (DAG) with 1,047,209 edges.

From this cleaned DAG, we extracted a cohort-specific subgraph by selecting all diagnosis nodes represented in our patient cohort. To improve connectivity and

contextual depth, we included the direct ancestors (up to two levels above) and direct descendants (up to two levels below) of each diagnosis node.

SNOMED CT vector embedding

We embedded the nodes of the cohort-specific subgraph into a 128-dimensional latent space using the node2vec algorithm implemented in the node2vec Python package (version 0.5.0). The following parameters were used: walk length = 20, number of walks = 20, return parameter $p = 3$, in-out parameter $q = 1.5$, and window size = 10. To ensure reproducibility, we fixed the random seed (seed = 42) and restricted the number of parallel workers to one. These settings generated stable and deterministic embeddings for downstream analysis.

SNOMED CT clustering

To cluster the embedded nodes, we tested three clustering methods: k-means, agglomerative, and spectral clustering. For each method, we evaluated a range of cluster numbers ($k = 10$ to 200) and computed the Silhouette score using cosine distance. A non-linear saturation model was then fitted to the Silhouette score curves using a beta-optimized asymptotic function. The optimal number of clusters was defined as the smallest k where the Silhouette score reached 97% of its fitted asymptote. Among the tested methods, k-means achieved the best performance with a Silhouette score of 0.31 at $k=43$, and was selected for downstream analysis.”

Minor Comments:

1. The authors used both urine and plasma proteome to extract clusters of signatures. How does this classification reflect on each patient? Do urine and plasma clusters co-occur? Meaning, are there patients/patient groups who based on one type of sample more similar, vs more dissimilar based on the other type of proteome?

We thank the reviewer for this important observation. We would like to clarify that the proteomic data itself does not feed into the patient clustering process. Patient grouping was performed entirely based on disease annotations using the SNOMED CT ontology, independent of the proteomics data. As a result, the assigned cluster for each patient is determined solely by their diagnosis and its hierarchical context, rather than by any molecular feature. Furthermore, not all patients contributed both plasma and urine samples, making direct patient-to-patient comparisons across the two fluids difficult to interpret.

That said, some patient groups do exhibit distinct proteomic signatures in either plasma or urine, which we have looked into mainly from the biomarker discovery angle. In our analysis, patients from several disease groups, including some of the more prevalent diseases, exhibited significantly altered proteins in one biofluid but not in the

other when compared to healthy controls. This observation indirectly reflects the applicability of said body fluid as a biomarker specimen for the respective disease.

To illustrate this, we compared the variance between the two body fluids across all shared clusters. In **Response Figure 3**, we show the median inter-individual coefficient of variation (CV) per cluster in urine versus PCA-N plasma. The analysis reveals that some clusters exhibit lower variance in plasma, while others are more stable in urine. These patterns may point to the relative informativeness or consistency of each fluid for different disease groups.

Response Figure 3. Comparison of median coefficient of variation (CV) per cluster between urine and PCA-N plasma datasets. Each point represents a shared cluster with its median CV in urine (x-axis) and PCA-N plasma (y-axis). The color scale indicates the ratio of sample sizes (urine / PCA-N plasma) for each cluster. The dashed diagonal line denotes identity ($y = x$), indicating equal CVs between urine and PCA-N plasma.

2. The authors used to obtain the ontology-based disease clusters. Have they compared other types of clustering? Louvain? Hierarchical classification based on the feature matrix, other types of community detection etc.

We thank the reviewers for suggesting alternative strategies. In this revised study, we now evaluated three different clustering approaches applied to the node2vec embeddings of the SNOMED CT disease subgraph: k-means clustering, agglomerative clustering, and spectral clustering. Each method was systematically assessed across a wide range of cluster numbers ($k = 10$ to 200), and the Silhouette

score was used to evaluate clustering quality. We selected the clustering result from k-means with $k = 43$, which achieved the highest Silhouette score (0.31), as the most robust solution for downstream analysis (Fig. EV5D–F).

We appreciate the suggestion to explore additional clustering approaches such as Louvain community detection, which could be applied directly on the native graph topology rather than the embedding space. However, in practice, Louvain and related clustering algorithms (e.g., Leiden) assigns each node to exactly one community. This constraint is not well suited to the polyhierarchical structure of SNOMED CT, where many clinical concepts have multiple parent terms and may simultaneously belong to different biological or clinical categories. In contrast, clustering in the node2vec-embedded space allows us to group semantically similar nodes while still preserving the richness of the hierarchical context.

3. "These clusters likely represent biologically meaningful groupings of related SNOMED CT terms, reflecting shared clinical or functional characteristics." It would be crucial to give some quantitative measure to assess clinical similarity within clusters.

We thank the reviewer for this helpful comment and fully agree that assessing the coherence of the disease clusters is essential and should be added. In this revision, we have incorporated several complementary metrics to evaluate cluster similarity and internal consistency.

First, we computed the Silhouette score for clustering solutions across multiple methods (k-means, agglomerative, spectral) and a range of cluster numbers ($k = 10$ to 200). The Silhouette score reflects how well each node fits within its assigned cluster based on its proximity to other nodes in the embedding space, balancing internal cohesion and separation from other clusters. Our final clustering solution using k-means at $k = 43$ achieved the highest Silhouette score (0.31), indicating that SNOMED CT disease terms within the same cluster exhibit strong semantic similarity in the latent space (Fig. EV5D–F).

Second, we computed pairwise Euclidean distances between disease nodes within each cluster to quantify internal cohesion. This provided an interpretable metric of how closely related the disease terms are in the embedding space (Fig. 4A).

Third, we performed a manual review of the diseases grouped within each cluster, guided by expert domain knowledge, to assess whether they reflect shared clinical or biological features (Table EV4).

These three levels of evaluation — structural (Silhouette score), geometric (Euclidean distance), and clinical (manual review) — together provide a robust assessment of the quality and interpretability of the clusters.

4. The authors highlight possible gaps in SNOMED ct ", reflecting a single patient and a single SNOMED CT term, likely corresponding to an extremely rare disease or a

limitation in the ontology's annotation. " How many of the clusters are single disease - disease annotation clusters? How many patients could not have been mapped to SNOMED IDs at all? How does SNOMED CT compare to other disease nomenclatures and classifications?

We thank the reviewer for bringing up this important point. In our cohort of 1,140 patients, all individuals were successfully mapped to at least one diagnosis with a corresponding SNOMED CT term, meaning there were no unmapped cases in terms of disease representation.

Regarding our clustering results, out of the 43 final SNOMED CT–based disease clusters, only two clusters consisted of a single disease node: Cluster 7 (Lyme arthritis) comprised of three patients and Cluster 22 (Gastritis caused by *Helicobacter pylori*) comprised of only one patient. These cases represent SNOMED CT terms that remained poorly connected even after graph expansion (i.e., inclusion of two levels of ancestor and descendant nodes). This likely reflects either the distinct clinical nature of the disease or limitations in the current SNOMED CT ontology in capturing its broader semantic relationships.

As for SNOMED CT itself, we chose this system because of its widespread use in electronic health records and its comprehensive hierarchical structure. Compared to other ontologies such as ICD-10 or Orphanet, SNOMED CT provides finer granularity and allows for polyhierarchical relationships (i.e., a term can belong to multiple parent categories). This makes it more suitable for the type of semantic graph embedding and clustering we performed. However, we acknowledge that no ontology is exhaustive. Some rare conditions are represented by only a single concept or have limited relationships to other terms, which may reduce their integration in cluster-based analyses.

28th Apr 2025

Dear Dr. Müller-Reif,

As you will see from their reports pasted below, while the referee #3 supports publication of the manuscript, referee #2 remains critical particularly regarding the clinical relevance. Based on the initial referee reports and after an editorial discussion, we agreed that the authors responded adequately to the referees' criticism and that the manuscript fits the scope of EMBO Molecular Medicine. Therefore, I am pleased to inform you that we will be able to accept your manuscript pending the following final amendments:

- 1) Please address all the referee #2 concerns. No additional experiments are required. Particular attention should be given to state the limitations of some experiments/conclusions and to tone down some conclusions where appropriate. Additionally, please implement referee #3 suggestion.
- 2) Figures: We note that some graphs are reused e.g. Fig. 3E and Fig. EV4C. Please cite in the respective figure legend every reused panel.
- 3) In the main manuscript file, please do the following:
 - Please address all comments suggested by our data editors listed below:
 - o Figure legends:
 1. Please indicate the statistical test used for data analysis in the legends of figures 3A, E; 4E, F; EV4 C, D.
 2. Please note that the box plots need to be defined in terms of minima, maxima, centre, bounds of box and whiskers, and percentile in the legends of figures 3D, 4C, D; EV1 D, E.
 3. Please note that the box plots need to be defined in terms of minima, maxima and percentile in the legends of figures 2A, B.
 4. Please note that information related to n is missing in the legends of figures 3B, C, D, F; 4C, D.
 - Remove all figures and only leave their legends at the end of the file.
 - Author contributions: Please remove it from the manuscript and specify author contributions in our submission system. CRediT has replaced the traditional author contributions section because it offers a systematic machine-readable author contributions format that allows for more effective research assessment. You are encouraged to use the free text boxes beneath each contributing author's name to add specific details on the author's contribution. More information is available in our guide to authors:
<https://www.embopress.org/page/journal/17574684/authorguide#authorshipguidelines>
 - Please remove Reagents and Tools Table and uploaded it as a separate file. Structured Methods section includes Reagents and Tools Table followed by a Methods and Protocols section. More information on how to adhere to this format as well as downloadable templates (.docx) for the Reagents and Tools Table can be found in our author guidelines:
<https://www.embopress.org/page/journal/17574684/authorguide#structuredmethods>
 - An example of a paper with Structured Methods can be found here:
<https://www.embopress.org/doi/full/10.1038/s44320-024-00037-6#sec-4>
 - In Methods, provide the statement that in addition to informed consent and WMA Declaration of Helsinki the experiments involving human subjects also conformed to the principles set out in the and the Department of Health and Human Services Belmont Report.
 - Indicate in legends exact n and exact p values, not a range, along with the statistical test used. To keep the figures "clear" some authors found providing an Appendix table Sx with all exact p-values preferable. You are welcome to do this if you want to.
 - Place "Data availability" before "Acknowledgments".
- 4) Tables: Please rename Tables EV1-EV4 to Datasets EV1-EV4 and Tables EV5 and EV6 to Tables EV1 and EV2 and update their callouts in the main manuscript file.
- 5) Funding: Please make sure that information about all sources of funding are complete in both our submission system and in the manuscript. Currently, the Charitable Helmsley Grant, the Care for Rare Foundation, and the Max Planck Society for Advancement of Science are missing in our system. Please correct.
- 6) The Paper Explained: Please provide "The Paper Explained" and add it to the main manuscript text. Please check "Author Guidelines" for more information. <https://www.embopress.org/page/journal/17574684/authorguide#researcharticleguide>
- 7) Synopsis: Every published paper now includes a 'Synopsis' to further enhance discoverability. Synopses are displayed on the journal webpage and are freely accessible to all readers. They include separate synopsis image and synopsis text.
 - Synopsis image: Please provide a visual abstract as a high-resolution jpeg file 550 px-wide x 200-600 pixels high to illustrate your article.
 - Synopsis text: Please provide a short standfirst (maximum of 300 characters, including space) as well as 2-5 one sentence bullet points that summarise the paper as a .doc file. Please write the bullet points to summarise the key NEW findings. They should be designed to be complementary to the abstract - i.e. not repeat the same text. We encourage inclusion of key acronyms and quantitative information (maximum of 30 words / bullet point). Please use the passive voice.
 - Please check your synopsis text and image before submission with your revised manuscript. Please be aware that in the proof stage minor corrections only are allowed (e.g., typos).
- 8) As part of the EMBO Publications transparent editorial process initiative (see our Editorial at <http://embomolmed.embopress.org/content/2/9/329>), EMBO Molecular Medicine will publish online a Review Process File (RPF) to accompany accepted manuscripts. This file will be published in conjunction with your paper and will include the anonymous

referee reports, your point-by-point response and all pertinent correspondence relating to the manuscript. Let us know whether you agree with the publication of the RPF and as here, if you want to remove or not any figures from it prior to publication. Please note that the Authors checklist will be published at the end of the RPF.

9) Please provide a point-by-point letter INCLUDING my comments as well as the reviewer's reports and your detailed responses (as Word file).

I look forward to reading a new revised version of your manuscript as soon as possible.

Yours sincerely,

Zeljko Durdevic

*** Instructions to submit your revised manuscript ***

- 1) a .docx formatted version of the manuscript text (including Figure legends and tables)
- 2) Separate figure files*
- 3) supplemental information as Expanded View and/or Appendix. Please carefully check the authors guidelines for formatting Expanded view and Appendix figures and tables at <https://www.embopress.org/page/journal/17574684/authorguide#expandedview>
- 4) a letter INCLUDING the reviewer's reports and your detailed responses to their comments (as Word file).
- 5) The paper explained: EMBO Molecular Medicine articles are accompanied by a summary of the articles to emphasize the major findings in the paper and their medical implications for the non-specialist reader. Please provide a draft summary of your article highlighting
 - the medical issue you are addressing,
 - the results obtained and
 - their clinical impact.This may be edited to ensure that readers understand the significance and context of the research. Please refer to any of our published articles for an example.
- 6) Author contributions: the contribution of every author must be detailed in a separate section.
- 7) EMBO Molecular Medicine now requires a complete author checklist (<https://www.embopress.org/page/journal/17574684/authorguide>) to be submitted with all revised manuscripts. Please use the checklist as guideline for the sort of information we need WITHIN the manuscript. The checklist should only be filled with page numbers where the information can be found. This is particularly important for animal reporting, antibody dilutions (missing) and

exact values and n that should be indicted instead of a range.

8) Every published paper now includes a 'Synopsis' to further enhance discoverability. Synopses are displayed on the journal webpage and are freely accessible to all readers. They include a short stand first (maximum of 300 characters, including space) as well as 2-5 one sentence bullet points that summarise the paper. Please write the bullet points to summarise the key NEW findings. They should be designed to be complementary to the abstract - i.e. not repeat the same text. We encourage inclusion of key acronyms and quantitative information (maximum of 30 words / bullet point). Please use the passive voice. Please attach these in a separate file or send them by email, we will incorporate them accordingly.

You are also welcome to suggest a striking image or visual abstract to illustrate your article. If you do please provide a jpeg file 550 px-wide x 300-600px high.

9) A Conflict of Interest statement should be provided in the main text

10) Please note that we now mandate that all corresponding authors list an ORCID digital identifier. This takes <90 seconds to complete. We encourage all authors to supply an ORCID identifier, which will be linked to their name for unambiguous name identification.

Currently, our records indicate that the ORCID for your account is 0000-0003-3454-2396.

Link Not Available

11) Include a Reagents and Tools Table as part of the Methods section, which can be downloaded from our author guidelines (<https://www.embopress.org/page/journal/17574684/authorguide#structuredmethods>)

Photos 400-800 DPI

*Additional important information regarding figures and illustrations can be found at

<https://bit.ly/EMBOPressFigurePreparationGuideline>. See also figure legend preparation guidelines:

<https://www.embopress.org/page/journal/17574684/authorguide#figureformat>

***** Reviewer's comments *****

Referee #2 (Remarks for Author):

While it is commendable that the analyses of dry have been significantly increased, the original lengthy paper has become even more extensive and therefore less readable. It is disappointing that the previous inquiry did not receive a clear response.

In Figure EV2, it is evident that there are very few instances of CV < 50 or CV < 20. This raises significant doubts about whether it is appropriate to make overall adjustments based on this data. Although the text emphasizes that numerous detections were achieved, it is crucial to maintain composure and evaluate based on properly measured data. Furthermore, in cases where renal disease overlaps and urinary protein levels are elevated, the findings may not be applicable. Instead of merely citing and utilizing the methods, it would be advisable to present corrected values using classic measures such as creatinine.

Precision medicine necessitates both precision and accuracy in measurements. It is likely that this technique will yield different types of data. My understanding is that precision can only be achieved with idealized measurement systems.

On Page 10, Lines 33-34, the assertions regarding the enrichment of keratinocyte differentiation and autocrine signaling are unclear in their meaning. Is there a possibility that these findings stem from sampling artifacts rather than pathological factors?

Regrettably, no new discoveries have emerged; rather, it appears to merely echo well-established findings. For example, while the importance of IgD changes in cystic fibrosis (CF) is mentioned, there are already published reports on this subject. Notably, IgD can be detected without requiring mass spectrometry, allowing for more accurate assessments.

There appear to be limitations concerning blood analysis. It is unclear how these findings can be applied clinically. The impressive nature of the techniques alone has not transcended the existing body of knowledge, which ultimately leads to the conclusion.

Contrary to the author's recognition, I have ultimately come to the realization that developing novel diagnostics through proteomics is challenging. The establishment of a precise measurement system for relevant molecules is essential for their use in diagnostics. It appears that the study remains focused solely on confirming existing measurement techniques and knowledge frameworks. While I believe this is a general observation, it is essential to ensure that readers who may not understand this concept are not misled; thus, a change in the title may be warranted.

Discussion

Page 27, Line 29: "Additionally, urine is uniquely sensitive to changes in the early stages of disease due to the lack of homeostatic mechanisms that buffer subtle proteomic shifts, 30 unlike plasma, which is tightly regulated."

This statement is likely an opinion; however, numerous exceptions exist, which renders it untrue. The description is too vague. It would be prudent to consult experts well-versed in renal tubular dynamics.

This stands in stark contradiction to findings already established in subsequent cardiovascular disease studies. While the foundational faculty may have provided the initial descriptions, discussions from a clinical perspective would likely be more beneficial.

Page 28, Line 10:

"This expanded proteomic coverage provided the resolution necessary to detect meaningful molecular signatures within the aggregated clusters, allowing us to draw clinically relevant conclusions from grouped rare diseases."

This assertion lacks verification and merely reiterates past reports. It remains unclear how this can be utilized clinically, and it appears to be an overdiscussion.

Referee #3 (Comments on Novelty/Model System for Author):

Itang et al have significantly expanded and improved their analysis by including multiple necessary comparisons, validation of their clustering description of methods and commenting on the potential of their approach to improve the analysis of rare diseases in a clinical and research setting. I am positive that this paper will have a high impact as a proof-of-concept of such approach.

Referee #3 (Remarks for Author):

Itang et al have significantly expanded and improved their analysis by including multiple necessary comparisons, validation of their clustering description of methods and commenting on the potential of their approach to improve the analysis of rare diseases in a clinical and research setting. I am positive that this paper will have a high impact as a proof-of-concept of such approach.

One small last comment: I applaud the motivation behind explaining the methods of analysis in detail in the main text of the manuscript, definitely helps with understanding the results. However I feel like a shorter, more succinct version of them would suffice included in the results section, with a more detailed explanation in the methods.

The authors addressed the remaining editorial issues.

Reviewer #2

Comment:

While it is commendable that the analyses of dry have been significantly increased, the original lengthy paper has become even more extensive and therefore less readable. It is disappointing that the previous inquiry did not receive a clear response.

Response:

We thank the reviewer for acknowledging the additional analyses we incorporated. We recognize that the increased length may have affected readability. In response, we have carefully revised the manuscript to improve clarity, streamline the narrative, and remove redundancies wherever possible. We have also ensured that all previously raised concerns are directly addressed in this revision and have made further efforts to clarify our responses point-by-point.

Comment:

In Figure EV2, it is evident that there are very few instances of $CV < 50$ or $CV < 20$. This raises significant doubts about whether it is appropriate to make overall adjustments based on this data. Although the text emphasizes that numerous detections were achieved, it is crucial to maintain composure and evaluate based on properly measured data. Furthermore, in cases where renal disease overlaps and urinary protein levels are elevated, the findings may not be applicable. Instead of merely citing and utilizing the methods, it would be advisable to present corrected values using classic measures such as creatinine.

Precision medicine necessitates both precision and accuracy in measurements. It is likely that this technique will yield different types of data. My understanding is that precision can only be achieved with idealized measurement systems.

Response:

We thank the reviewer for this critical observation. We would like to clarify that Figure EV2 shows biological variability across individuals (inter-individual CVs), not analytical variability of the proteomic measurements. Analytical precision was evaluated separately using the reference channel of the mDIA setup, yielding a median analytical CV of 21% across quantified proteins, as stated in the "*Proteome data quality assessment*" section of the Results. This confirms that our proteomic measurements are sufficiently precise to reliably capture biological differences across individuals.

High biological (inter-individual) CVs in urinary proteomics are well-documented in the literature. For example, Virreira Winter et al. (EMBO Mol Med, 2021) similarly reported that most urinary proteins exhibit CVs greater than 20% across individuals. The inter-

individual variability observed in our study is consistent with previous large-scale urinary proteomics cohorts, while often other studies disregard such quality parameter at all and focus solely on differentially regulated proteins.

Regarding normalization, we standardized sample input by injecting equal amounts of total urinary protein (100 ng of peptides per sample) into the mass spectrometer. This total protein-based normalization approach is appropriate for discovery-based proteomic studies, where consistent technical input is critical for comparative analyses. While creatinine-based normalization is widely used in clinical diagnostic settings to correct for urine concentration variability, it was not applied here, because the peptide amount-based normalization strategy directly controls for differences in total protein input. Additionally, we applied label-free quantification normalization using directlfq (Ammar et al., MCP 2023) within each LC-MS run to control for minor injection variability.

Taken together, our workflow ensures that technical variability is minimized and that observed differences predominantly reflect true biological effects on the relative urine protein level per patient.

Comment:

On Page 10, Lines 33-34, the assertions regarding the enrichment of keratinocyte differentiation and autocrine signaling are unclear in their meaning. Is there a possibility that these findings stem from sampling artifacts rather than pathological factors?

Response:

We thank the reviewer for raising this important point regarding the enrichment of keratinocyte differentiation and autocrine signaling pathways across age in both males and females. In response, we revisited the specific proteins contributing to these enrichments within module 1.

For the GO term related to “keratinocyte differentiation”, the main contributing proteins (TGM1, TGM3, IVL, SPRR1B, SPRR3, DSP, ANXA1, EVPL) are well-established markers of skin keratinocyte differentiation and barrier formation (Candi *et al*, 2005; Chermnykh *et al*, 2020; Surbek *et al*, 2023). The reviewer rightly raises the possibility that these proteins could reflect sample contamination, for example through skin shedding during urine collection. While we cannot fully rule out minor contributions from epithelial material, we believe that the overall pattern is better explained by biology.

The development and maturation of the skin barrier is a hormonally regulated process that accelerates during puberty. During this period, levels of sex hormones, including androgens and estrogens, rise sharply and act directly on skin cells. Androgens stimulate keratinocyte proliferation and sebaceous gland activity, while estrogens influence epidermal differentiation and barrier integrity. These hormonal effects drive the remodeling and functional maturation of the skin barrier during adolescence, with differences in timing

and magnitude between males and females (Zouboulis & Degitz, 2004; Zouboulis *et al*, 2007; Zouboulis, 2009).

Importantly, module 1 was not associated with any specific disease group but showed a clear, age-dependent trajectory across the cohort, with a sharper increase in females after age 9 compared to males. Given the randomized processing of samples, this structured pattern is unlikely to result from sampling artifacts and is more consistent with coordinated physiological changes during puberty.

For the GO term associated with “autocrine signaling”, the contributing proteins (SERPINB3 and S100A9) are involved in epithelial homeostasis, immune modulation, and tissue remodeling. Both proteins are expressed in epithelial tissues and can be influenced by hormonal and immune changes during puberty, further supporting their biological relevance in this context.

Comment:

Regrettably, no new discoveries have emerged; rather, it appears to merely echo well-established findings. For example, while the importance of IgD changes in cystic fibrosis (CF) is mentioned, there are already published reports on this subject. Notably, IgD can be detected without requiring mass spectrometry, allowing for more accurate assessments.

Response:

We thank the reviewer for this comment. We agree that changes in IgD levels in cystic fibrosis have been reported previously, and we did not intend to present this as a novel finding. Instead, the fact that our clustering approach captures such known biology supports the validity of the approach. This study is meant as a proof-of-concept of population proteomics in the pediatric realm and how to approach analysis of rare and in general underrepresented subpopulations. We have clarified this framing in the revised manuscript.

Comment:

There appear to be limitations concerning blood analysis. It is unclear how these findings can be applied clinically. The impressive nature of the techniques alone has not transcended the existing body of knowledge, which ultimately leads to the conclusion.

Contrary to the author's recognition, I have ultimately come to the realization that developing novel diagnostics through proteomics is challenging. The establishment of a precise measurement system for relevant molecules is essential for their use in diagnostics. It appears that the study remains focused solely on confirming existing measurement techniques and knowledge frameworks. While I believe this is a general observation, it is essential to ensure that readers who may not understand this concept are not misled; thus, a change in the title may be warranted.

Response:

We agree that clinical applicability requires targeted validation, which is beyond the scope of this study. Our aim here was to demonstrate that an ontology-guided clustering approach can reveal biologically meaningful proteomic patterns across rare pediatric diseases, using a standardized, discovery-based workflow. This work is intended as a systems-level proof-of-concept to demonstrate the value of the approach, not as a step toward immediate clinical application. Any future diagnostic use would require follow-up studies with targeted assays specifically designed for precision and clinical validation.

That said, we believe that the current title accurately reflects the scope of the study, as it does not claim the identification of a biomarker or the development of a diagnostic tool, but rather highlights the use of an ontology-guided clustering strategy to explore proteomic variation across rare pediatric disorders.

Comment:

Discussion

Page 27, Line 29: "Additionally, urine is uniquely sensitive to changes in the early stages of disease due to the lack of homeostatic mechanisms that buffer subtle proteomic shifts, 30 unlike plasma, which is tightly regulated."

This statement is likely an opinion; however, numerous exceptions exist, which renders it untrue. The description is too vague. It would be prudent to consult experts well-versed in renal tubular dynamics.

This stands in stark contradiction to findings already established in subsequent cardiovascular disease studies. While the foundational faculty may have provided the initial descriptions, discussions from a clinical perspective would likely be more beneficial.

Response:

We thank the reviewer for this comment. The original statement was based on literature suggesting that urine may reflect early molecular changes in tissue-proximal conditions due to the lack of systemic homeostatic buffering (Zhang et al., 2022). However, we agree that this does not apply across all diseases. In particular, early detection of cardiovascular conditions relies heavily on plasma biomarkers such as hsCRP, IL-6, and troponins, which are directly released into circulation (Ridker et al., 2004; Wu, 2019). We have revised the manuscript to reflect this disease- and matrix-specific sensitivity more accurately (p. 14, lines 29-41).

"Additionally, urine is uniquely sensitive to changes in the early stages of disease, particularly in renal and urinary tract conditions, due to its limited systemic buffering (Zhang et al., 2022). However, this sensitivity is disease- and tissue-specific. In systemic conditions such as cardiovascular disease, plasma biomarkers such as hsCRP, interleukins, and troponins remain the gold standard for early detection, owing to their direct release into circulation (Ridker *et al*, 2004; Kukova *et al*, 2019). These distinctions highlight the complementary roles of plasma and urine: while plasma

reflects systemic inflammation and immune responses, urine captures tissue-localized changes that may otherwise be missed. Analyzing both fluids in parallel provides a more comprehensive understanding of disease biology.”

Comment:

Page 28, Line 10:

"This expanded proteomic coverage provided the resolution necessary to detect meaningful molecular signatures within the aggregated clusters, allowing us to draw clinically relevant conclusions from grouped rare diseases."

This assertion lacks verification and merely reiterates past reports. It remains unclear how this can be utilized clinically, and it appears to be an overdiscussion.

Response:

We thank the reviewer for this comment and acknowledge that the original phrasing may have implied more clinical relevance than intended. In the context of discovery, our goal was to identify biologically meaningful patterns across disease clusters. With the expanded proteomic coverage afforded by the PCA-N workflow, we detected approximately twice as many different proteins between aggregated disease clusters and healthy controls compared to the neat plasma workflow. These additional hits may reflect shared disease biology and warrant further investigation. We have revised the manuscript to better reflect this intent (p. 15, lines 12-17).

“This expanded proteomic coverage provided the resolution necessary to detect meaningful molecular signatures within the aggregated clusters, allowing us to uncover shared biological features across grouped rare diseases. These findings highlight the synergistic potential of combining deep molecular data from proteomics with clinically informed ontological structures in discovery-driven research.”

Reviewer #3:

Comment:

Itang et al have significantly expanded and improved their analysis by including multiple necessary comparisons, validation of their clustering description of methods and commenting on the potential of their approach to improve the analysis of rare diseases in a clinical and research setting. I am positive that this paper will have a high impact as a proof-of-concept of such approach.

One small last comment: I applaud the motivation behind explaining the methods of analysis in detail in the main text of the manuscript, definitely helps with understanding the results. However, I feel like a shorter, more succinct version of them would suffice included in the results section, with a more detailed explanation in the methods.

Response:

We thank the reviewer for their positive and encouraging assessment of our work. We are especially grateful for the recognition of our proof-of-concept approach and its potential to impact rare disease research. In response to the reviewer's suggestion, we have revised the Results section to streamline the methodological explanations and moved the more detailed technical descriptions into the Methods section, where appropriate. We hope this improves readability without sacrificing clarity or transparency.

9th May 2025

Dear Dr. Müller-Reif,

We are pleased to inform you that your manuscript is accepted for publication and is now being sent to our publisher to be included in the next available issue of EMBO Molecular Medicine. Please make sure that all data deposited in public repositories are freely accessible upon publication.
